# Investigation of several proxies to estimate sulfuric acid concentration in volcanic plume conditions

Clémence Rose[1], Matti P. Rissanen[2,3], Siddharth Iyer[2], Jonathan Duplissy[3,4], Chao Yan[3], John B. Nowak[5], Aurélie Colomb[1], Régis Dupuy[1], Xu-Cheng He[3], Janne Lampilahti[3], Yee Jun Tham[3], Daniela Wimmer[3], Jean-Marc Metzger[6], Pierre Tulet[7], Jérôme Brioude[7], Céline Planche[1], Markku Kulmala[3] and Karine Sellegri[1]

[1]Université Clermont Auvergne, CNRS, Laboratoire de Météorologie Physique (LaMP), F-63000 Clermont-Ferrand, France
[2]Aerosol Physics Laboratory, Physics Unit, Tampere University, Tampere, Finland
[3]Institute for Atmospheric and Earth System Research INAR/Physics, Faculty of Science, University of Helsinki, Helsinki, Finland
[4]Helsinki Institute of Physics, University of Helsinki, Helsinki, Finland
[5]Chemistry and Dynamics Branch, NASA Langley Research Center, Hampton, VA 23681, USA
[6]Observatoire des Sciences de l'Univers de La Réunion, UMS 3365 (CNRS, Université de La Réunion, Météo-France), 97744 Saint Denis de La Réunion, France
[7]LACy, Laboratoire de l'Atmosphère et des Cyclones, UMR8105 (CNRS, Université de La Réunion, Météo-France), Saint-Denis de la Réunion, France

*Correspondence to*: c.rose@opgc.fr

**Abstract**

Sulfuric acid ($H_2SO_4$) is commonly accepted as a key precursor for atmospheric new particle formation (NPF). However, direct measurements of $[H_2SO_4]$ remain challenging, thus preventing the determination of this important quantity, and, consequently, a complete understanding of its contribution to the NPF process. Several proxies have been developed to bridge the gaps, but their ability to predict $[H_2SO_4]$ in very specific conditions such as those encountered in volcanic plumes (including in particular high sulphur dioxide mixing ratios) has not been evaluated so far. In this context, the main objective of the present study was to develop new proxies for daytime $[H_2SO_4]$ in volcanic plume conditions and compare their performance to that of the proxies available in the literature. In specific, the data collected at Maïdo during the OCTAVE 2018 campaign, in the volcanic eruption plume of the Piton de la Fournaise, were first used to derive seven proxies based on the knowledge of sulphur dioxide ($SO_2$) mixing ratio, global radiation, condensation sink (CS) and relative humidity (RH). A specific combination of some or all of these variables was tested in each of the seven proxies. In three of them (F1-F3), all considered variables were given equal weight in the prediction of $[H_2SO_4]$, while adjusted powers were allowed (and determined during the fitting procedure) for the different variables in the other four proxies (A1-A4). Proxies A1-A4 were overall found to perform better compared to F1-F3, with, in specific, improved predictive ability for $[H_2SO_4] > 2\times10^8$ cm$^{-3}$. The CS was observed to play an important role in regulating $[H_2SO_4]$, while, in contrast, the inclusion of RH did not improve the predictions. A last expression accounting for an additional sink term related to cluster formation, S1, was also tested and showed a very good predictive ability over the

whole range of measured [H$_2$SO$_4$]. The newly developed proxies were in a second step further evaluated using airborne measurements performed in the passive degassing plume of Etna during the STRAP 2016 campaign. Increased correlations between observed and predicted [H$_2$SO$_4$] were obtained when the dependence of predicted [H$_2$SO$_4$] over CS was the lowest, and when the dependence over [SO$_2$] was concurrently the highest. The best predictions were finally retrieved by the simple

formulation of F2 (in which [SO$_2$] and radiation alone were assumed to explain the variations of [H$_2$SO$_4$] with equal contributions), with a pre factor adapted to the STRAP data. All in all, our results illustrate the fairly good capacity of the proxy available in the literature to describe [H$_2$SO$_4$] in volcanic plume conditions, but highlight at the same time the benefit of the newly developed proxies for the prediction of the highest concentrations ([H$_2$SO$_4$] > 2-3×10$^8$ cm$^{-3}$). Also, the contrasting behaviours of the new proxies in the two investigated datasets indicate that in volcanic plumes like in other environments, the

relevance of a proxy can be affected by changes in environmental conditions, and that location specific coefficients do logically improve the predictions.

## 1 Introduction

Sulfuric acid (H$_2$SO$_4$) is commonly accepted as a key precursor for atmospheric new particle formation (NPF), and is therefore considered as the main driving species of cluster formation in air quality and climate models (Semeniuk et al., 2018). While it

may not always be determinant in the process (Kirkby et al., 2016; Rose et al., 2018), evidence for the contribution of H$_2$SO$_4$ to the formation and initial growth of particles was reported from chamber experiments (Kirkby et al., 2011; Duplissy et al., 2016; Lehtipalo et al., 2018) and atmospheric measurements performed in various environments (Frege et al., 2017; Jokinen et al., 2018; Yan et al., 2018). The overall predominant role played by sulfuric acid in cluster formation is in particular related to its very low saturation vapour pressure and high hydrogen bonding capacity (Zhang et al. 2011). Based on laboratory studies,

H$_2$SO$_4$ participates in different nucleation mechanisms, including binary water – sulfuric acid cluster formation (*e.g.* Kirkby et al., 2011; Duplissy et al., 2016) and ternary formation pathways, in which a third stabilizing species, either inorganic (e.g. Jen et al., 2014; Kürten et al., 2016) or organic (*e.g.* Zhang et al., 2004; Riccobono et al., 2014), is involved. The relative importance of these mechanisms is expected to vary spatially, both horizontally and vertically, depending on environmental conditions. These, including the availability of precursors as well as temperature and relative humidity levels, also influence

the contribution of ions in the process, which seems to be overall limited in the planetary boundary layer, but could in contrast be more significant in the middle and upper troposphere (*e.g.* Lovejoy et al., 2004; Hirsikko et al., 2011; Duplissy et al. 2016). Recently, NPF was observed in the volcanic eruption plume of the Piton de la Fournaise (La Réunion Island) (Rose et al., 2019) and in the passive degassing plumes of Etna and Stromboli (Italy) (Sahyoun et al., 2019), where a tight connection between the particle formation rate and [H$_2$SO$_4$] was highlighted. Using indirect quantification of [H$_2$SO$_4$], Rose et al. (2019)

concurrently reported a fair agreement between measured cluster formation rates and the values derived from the recent parameterization of water – sulfuric acid binary nucleation by Määttänen et al. (2018), which also predicted a likely significant contribution of ion-induced nucleation in the volcanic plume for [H$_2$SO$_4$] below ∼ 8 × 10$^8$ cm$^{-3}$.

The identification of the vapours involved in the first stages of NPF and further understanding of the process requires a direct characterization of the relevant clusters and their precursors. Information about the species contributing to cluster formation

with sulphuric acid and preferential formation pathways was gained from laboratory studies (Hanson et al., 2002, 2006). Laboratory experiments have also made it possible to evaluate instrumental setups and related protocols for accurate detection and quantification of the clusters and their precursors (Jen et al., 2016; Riva et al., 2019). Measurements of $[H_2SO_4]$ are currently performed with nitrate ion ($NO_3^-$) based chemical ionization mass spectrometers (CIMS). In specific, the Atmospheric Pressure interface Time Of Flight mass spectrometer (APi-TOF, Aerodyne Research Inc. and TOFWERK AG;

Junninen et al., 2010) equipped with a chemical ionization (CI) inlet (CI-APi-TOF, Jokinen et al., 2012) has been used in most of the recent studies, as it offers a detection limit which is low enough ($\sim 2\times 10^4$ cm$^{-3}$) to allow measurements of $[H_2SO_4]$ in typical daytime atmospheric conditions ($10^5 - 10^7$ cm$^{-3}$) (Kirkby et al., 2016). In the study of Sahyoun et al. (2019), the CI inlet was replaced by an ambient ionization (AI) inlet specifically developed to meet the safety regulation requirements regarding the use of chemicals on-board the French ATR-42 research aircraft.

However, as recently noticed by Lu et al. (2019), direct measurements of $[H_2SO_4]$ remain challenging, because the deployment of CIMS and the analysis of the data they provide require specific expertise. Therefore, for studies in which $[H_2SO_4]$ is an important variable (*i.e.* mainly for nucleation and NPF studies), is it useful to be able to predict it from more accessible observations such as $SO_2$ concentration and environmental parameters. This is why several proxies for $[H_2SO_4]$ have been developed, based on the assumption that $H_2SO_4$ mostly results from the oxidation of the sulphur dioxide ($SO_2$) by the hydroxyl

radical (OH). These proxies thus neglect the possible contribution of oxidants other than OH, including for instance stabilized Criegee Intermediates (Mauldin et al., 2012) or other proposed compounds (*e.g.* halogen oxides, Berresheim et al., 2014), and hence they only predict daytime $[H_2SO_4]$. The first proxies for $[H_2SO_4]$ were derived by Petäjä et al. (2009) from measurements performed at the SMEAR II station, in the boreal forest. In line with the theory, Petäjä and co-workers simply expressed $[H_2SO_4]$ as the ratio of a source term ($[SO_2]\times[OH]$, with [OH] possibly replaced by UVB or global radiation intensity due to

difficulty in obtaining atmospheric [OH]) over a sink term (related to condensation), both having the same weight in the proxy (*i.e.* powers equal to unity). Shortly after, Mikkonen et al. (2011) provided new proxies derived from measurements collected at different sites representative of contrasting environments in Europe and North America, with also more variables, including the temperature dependent reaction rate between $SO_2$ and OH as well as relative humidity (RH). In addition, individual powers were allowed for the different variables, and defined by applying a nonlinear least squares fitting to the datasets collected at

each site, and, finally, to the combined dataset. Lu et al. (2019) followed a similar approach and developed proxies based on measurements performed in the urban area of Beijing, some of which included $[O_3]$ and [HONO] due to the contribution of these two compounds in the production of OH radicals. Main motivation for the study of Lu and co-workers was the likely limited relevance of the previous proxies for the study of frequently highly polluted atmospheres such as those of Chinese megacities. More recently, Dada et al. (2020) derived new expressions to account for the production of $H_2SO_4$ via the oxidation

of SO$_2$ by stabilized Criegee Intermediates and for the loss of H$_2$SO$_4$ due to cluster formation. This additional source term interestingly made the prediction of night time H$_2$SO$_4$ possible, and overall, contributed to improve, together with the cluster term, the predictive ability of the proxies also during daytime. However, one limitation to the use of such formulation is that it requires information on alkenes concentration which is often not available, as pointed out by Dada et al. (2020).

In absence of direct measurements, and also of a specific proxy, Boulon et al. (2011) and Rose et al. (2019) used the expression from Petäjä et al. (2009) and Mikkonen et al. (2011), respectively, to estimate [H$_2$SO$_4$] and evaluate its connection with NPF in the volcanic eruption plumes of the Eyjafjallajokull and the Piton de la Fournaise, respectively. However, the lack of measured [H$_2$SO$_4$] obviously did not make it possible in these studies to assess the performance of the abovementioned proxies in such unusual conditions, which has motivated the present work. The main objective of this study was to develop new proxies

for [H$_2$SO$_4$] in volcanic plume conditions and compare their predictive ability to that of the proxies available in the literature. For that purpose, the first direct measurements of [H$_2$SO$_4$] conducted in plume conditions in the frame of two different projects were used. In specific, the data collected at the Maïdo observatory (La Réunion Island) during the OCTAVE campaign, in the volcanic eruption plume of the Piton de la Fournaise, were used to derive the proxies, and their performance were in a second step further evaluated using airborne measurements performed in the passive degassing plume of Etna during the STRAP

campaign (Sahyoun et al., 2019). We otherwise followed the same approach as Mikkonen et al. (2011) and Dada et al. (2020) to develop the proxies.

## 2. Measurements

### 2.1 Ground-based measurements performed at the Maido observatory during OCTAVE campaign

Measurements were performed at the Maïdo observatory located on La Réunion Island, in the Indian Ocean (21.080° S, 55.383°

E, 2150 m a.s.l.) in the frame of the OCTAVE 2018 (Oxygenated Compounds in the Tropical Atmosphere: Variability and Exchange, http://octave.aeronomie.be) campaign (Fig. 1), which took place between March 7th and May 8th 2018 (note that for simplicity, this dataset will be hereafter referred to as OCTAVE). Specific attention was paid to the time period between April 28th and May 4th, during which the volcanic eruption plume of the Piton de la Fournaise, located ~ 39 km away from Maïdo in the south-eastern region of the island, was detected at the station (see Section 2.3). Detailed information about the facility can

be found in Baray et al. (2013), and the instrumental setup used in the present study was, to a large extent, already described in Foucart et al. (2018) and Rose et al. (2019). Briefly, the aerosol number size distribution between 10 and 600 nm was retrieved every 7 min by a custom-built differential mobility analyser (DMPS) operated behind a whole air inlet (higher size cut-off of 25 µm for an average wind speed of 4 m s$^{-1}$). These measurements were used to calculate the condensation sink (CS), which represents the loss rate of vapours, and in specific H$_2$SO$_4$, on pre-existing particles (Kulmala et al., 2012). In

contrast with the abovementioned studies, SO$_2$ mixing ratios were measured every minute with a Teledyne API T100U analyser, which has a lower detection limit (0.05 ppb) than that of the instrument previously running at the site. In addition, O$_3$ and NO$_x$ mixing ratios were respectively measured with a Thermo Scientific 49i analyser and a Teledyne API T200UP

analyser, with detection limits of 1 and 0.05 ppb, respectively, and a time resolution of 1 minute. Like in Foucart et al. (2018) and Rose et al. (2019), meteorological parameters measured with a time resolution of 3 s were used, including global radiation (SPN1, DeltaT Devices Ltd., resolution 0.6 Wm$^{-2}$), temperature and RH (Vaisala Weather Transmitter WXT510).

A CI-APi-ToF mass spectrometer employing nitrate reagent ions was deployed to retrieve gas-phase $H_2SO_4$ as detailed in Jokinen et al. (2012). Briefly, sulfuric acid is deprotonated by $NO_3^-$ ions in the chemical ionization inlet, and then directly quantified from signals of resulting bisulfate ion ($HSO_4^-$) and its nitric acid cluster ($HNO_3 \cdot HSO_4^-$). At the highest ambient concentrations, also the bisulfate clusters with sulfuric acid and sulfuric acid dimer (*i.e.*, $H_2SO_4 \cdot HSO_4^-$ and $(H_2SO_4)_2 \cdot HSO_4^-$, respectively) made a non-negligible contribution, and were included in the analysis. To convert the measured ion signals into concentrations, Eq. (1) was used. The measured raw ion signals were normalized by the reagent ion current and multiplied by a calibration factor of $C = 1.7 \times 10^{10}$ molecule cm$^{-3}$, which was determined by a procedure outlined in Kürten et al. (2012):

$$[H_2SO_4] = \frac{HSO_4^- + H_2SO_4 \cdot HSO_4^- + (H_2SO_4)_2 \cdot HSO_4^- + H_2SO_4 \cdot NO_3^-}{NO_3^- + HNO_3 \cdot NO_3^- + (HNO_3)_2 \cdot NO_3^-} \times C \tag{1}$$

Note that the mass spectrometer was calibrated onsite, in the exact position it was sampling the ambient air during the measurement campaign, and up to the high sulfuric acid concentrations observed under the plume conditions.

## 2.2 Airborne measurements performed during the STRAP campaign

Airborne measurements were conducted in the frame of the STRAP campaign, on May 15[th] and 16[th] 2016, in the volcanic plumes of Etna and Stromboli, during passive degassing (Pianezze et al., 2019; Sahyoun et al., 2019). In the present work, we focussed on flight ETNA 13 performed around Etna on May 15[th], with specific attention to the first part of the flight between 10:43 and 11:00 UTC (LT -2h). This specific period was selected as measurements were performed at constant altitude (~ 2900 m), thus making the overall interpretation of the observations easier, and, more importantly, because several latitudinal plume transects were performed at distances between ~ 7 and 39 km from the vent, resulting in very clear variations of $[H_2SO_4]$ (see Fig. 2). For simplicity, while being restricted to the first part of flight ETNA 13, this data will nonetheless be referred to as STRAP data hereafter.

Measurements were performed on-board the French ATR-42 research aircraft operated by SAFIRE (Service des Avions Français Instrumentés pour la Recherche en Environnement). The instrumental setup available on-board the aircraft was previously described by Sahyoun et al. (2019) and is only briefly recalled here. Particle number size distributions in the range between 90 and 3000 nm were measured with a Passive Cavity Aerosol Spectrometer Probe (PCASP-100X V3.11.0) and used for the calculation of the CS. One should keep in mind that the CS reported in the present work, while being already increased compared to that derived from > 250 nm particle number concentration by Sahyoun et al. (2019), most likely remains a lower estimate of the actual sink. Indeed, the contribution of sub-90 nm particles to the condensational sink was expected to be significant, in specific because new particle formation and growth events were observed to occur in the passive degassing plume of Etna during flight ETNA 13 (Sahyoun et al., 2019, Fig. 3.a). $SO_2$, $O_3$ and $NO_x$ mixing ratios were measured with analysers of the exact same type as those used at Maïdo, which main characteristics are recalled in the previous section. Routine

meteorological parameters, together with geographical parameters, which are continuously monitored on-board the aircraft, were in addition used in the analysis, including global radiation, temperature, RH, pressure, altitude, latitude and longitude. All measurements listed above were retrieved with a 1 s time resolution.

As previously explained in details by Sahyoun et al. (2019), sulfuric acid concentrations were measured with an APi-TOF

equipped with an ambient ionization (AI) inlet adapted to airborne measurements and used for the first time during STRAP. In contrast with the CI inlet, the AI inlet does not require the use of any chemicals, and only includes a soft X-ray source (Hamamatsu L9490) to ionize the sample flow. This direct ionization process was sufficient to get a high enough signal and allow a time resolution as high as 1s for the corresponding measurements. Also, in order to avoid possible effects related to pressure changes on the detection of the AI-Api-TOF, a pressure stabilizing unit was installed in front of the instrument. As

detailed in Sahyoun et al. (2019), calibration of this new setup was performed (with respect to [$H_2SO_4$] measurement) during fall 2016 at the CLOUD CERN facility (Kirkby et al., 2011; Duplissy et al., 2016 and references therein) by comparison with the measurements performed with a nitrate based CI-APi-TOF in various conditions representative of the atmosphere. During these experiments, $O_2^-$ was assumed to be the main ionizing agent of $H_2SO_4$, as on board the aircraft during the measurement campaign, but contribution of $NO_3^-$ could not be excluded, in particular in presence of higher NOx levels (up to 33 ppb) in the

CLOUD chamber. Therefore, estimates of [$H_2SO_4$] were finally obtained by the mean of Eq. (2) using the signals measured at $m/z = 97$ Th ($HSO_4^-$) and $m/z = 160$ Th ($NO_3^- \cdot H_2SO_4$) by the AI-APi-TOF and a calibration coefficient $C = 4.5 \times 10^9$ molecule $cm^{-3}$:

$$[H_2SO_4] = \frac{HSO_4^- + NO_3^- \cdot H_2SO_4}{Total\ ion\ count} \times C \tag{2}$$

The good correlation obtained between the signals of the well characterized CI-APi-TOF and the AI-APi-TOF during the

calibration experiments can undoubtedly be seen as an indicator of the satisfactory performance of the newly developed inlet, and further on the derivation of [$H_2SO_4$] (see Fig. S3 in the Supplement of Sayhoun et al. 2019). However, it cannot be excluded that [$H_2SO_4$] inferred from the measurements carried out during the STRAP campaign were subject to greater uncertainty due to the specific conditions of the volcanic plume, in particular with respect to $H_2SO_4$ concentrations, which were on average slightly higher in the plume than in the simulation chamber ($< \sim 5 \times 10^7$ $cm^{-3}$ in the CLOUD chamber vs $\sim 1.6 \times 10^8$ $cm^{-3}$ on

average during the flight segment of interest, see Table 1 and Fig. 2).

## 2.3 Overview of the two campaign datasets

An overview of the campaign conditions is presented in Fig. 1 for OCTAVE, and, similarly, in Fig. 2 for the first part of flight ETNA 13 performed during STRAP. A broader view of the STRAP campaign, including time series of the variables of interest during all flights, can in addition be found in Sahyoun et al. (2019). For consistency with the previous studies by Rose et al.

(2019) and Sahyoun et al. (2019), the occurrence of volcanic plume conditions was assessed based on $SO_2$ mixing ratios and a threshold of 2 ppb was used for the detection of the plume for both datasets. Daytime in-plume conditions identified during OCTAVE are highlighted by the red boxes on Fig. 1. Specific attention was paid to daytime, when global radiation exceeded

10 W m$^{-2}$, as the proxies for [H$_2$SO$_4$] discussed in the next sections were assumed to only apply in these conditions, following earlier work by Mikkonen et al. (2011). Note that the eruptive volcanic plume of the Piton de la Fournaise was also detected at Maïdo after May 4$^{th}$, but only during sporadic events, which were thus excluded from the present study. Concerning STRAP, measurements were mostly collected in plume conditions during the selected period, as illustrated on Fig. 2a. In-plume

conditions were only interrupted during a short 1 min period at ~10h51 UTC, when the aircraft exited the volcanic plume at the end of a latitudinal transect, and during the last 30 seconds of the selected period, for the same reason (Fig. 2b). Moreover, gaps in radiation data were caused by improper measurements during turns, when the aircraft itself affected the amount of radiation reaching the sensor. In addition to Fig. 1 and 2, median and 5$^{th}$-95$^{th}$ percentiles of a number of key atmospheric variables are reported in Table 1. Statistics derived from measurements performed under regular conditions at Maïdo (*i.e.*

outside of the volcanic plume) are reported as well for further investigation of the influence of the volcanic plume on atmospheric conditions at the site. Note that this "reference" period (Fig. 1, green boxes) was limited to April 18$^{th}$ due to the occurrence of a tropical cyclone on the next days, which had a noticeable effect on atmospheric mixing (see for instance temperature) and lead as well to measurement issues (*e.g.* DMPS). Data collected before April 11$^{th}$ were on the other hand excluded from the present work as H$_2$SO$_4$ measurements could not be performed until this date.

As evidenced in Table 1 and Figs. 1 and 2, in-plume meteorological conditions logically showed stronger variability during OCTAVE than during STRAP, as a result of the longer investigated period and, more importantly, because continuous measurements performed during OCTAVE allowed for the capture of diurnal cycles. Nonetheless, the medians derived from the two datasets were closely located for a number of variables, with, in specific, a ratio < 1.5 for temperature, RH, CS as well as for SO$_2$, O$_3$ and NO$_x$ mixing ratios. Note that O$_3$ and NO$_x$ mixing ratios were only reported for further comparison of the

two campaign datasets and investigation of the conditions in and out of plume; excepted an attempt to include [O$_3$] in a sensitivity test, they were not used in the derivation of the proxies. In contrast, larger differences were observed for global radiation and [H$_2$SO$_4$]. As mentioned earlier, lower median radiation in OCTAVE, calculated from all data > 10 Wm$^{-2}$, reflects the diurnal cycle observed at the site, while measurements in STRAP were performed over a relatively short period during daytime, under clear sky conditions. Observed differences in median [H$_2$SO$_4$] were also likely related to campaigns design and

duration, but we could not exclude that part of this difference was also explained by the use of different instrumental setups in the two campaigns, and related as well to the uncertainty which is associated to the derivation of [H$_2$SO$_4$] from mass spectrometry data.

As previously reported by Rose et al. (2019), the comparison of in and out of plume conditions at Maïdo did not highlight any clear influence of the volcanic plume on meteorological conditions. O$_3$ levels were also very comparable, and median NO$_x$

level was observed to be slightly higher outside the plume (factor of 1.8). In contrast, SO$_2$ mixing ratios and [H$_2$SO$_4$] were logically higher in plume conditions, with one order of magnitude higher medians compared to regular conditions. Similar observation was also made for the CS, as a likely result of enhanced secondary aerosol formation in the volcanic plume (Rose et al., 2019). Compared with the sites in the studies by Mikkonen et al. (2011) and Lu et al. (2019), in-plume conditions were characterized by 1-2 and 1-3 orders of magnitude higher SO$_2$ mixing ratios and [H$_2$SO$_4$] medians, respectively, but comparable

or even lower CS. In specific, [$H_2SO_4$] measured in plume conditions were, to our knowledge, the highest ever recorded in the atmosphere. They could furthermore be lower estimations of the actual concentrations when approaching $10^9$ cm$^{-3}$, due to the likely occurrence of homogeneous condensation of $H_2SO_4$ above this threshold (Brus et al., 2010). These specificities of the volcanic plume clearly illustrate the need for 1) a deeper investigation of the performance, in these conditions, of the proxies available in the literature and 2) further development of proxies adapted to these very specific conditions.

Table 1 Median and $5^{th}$ – $95^{th}$ percentiles of key atmospheric variables measured during daytime in regular and plume conditions during OCTAVE, and in plume conditions during the first part of flight ETNA 13 (STRAP).

| | OCTAVE regular conditions | | OCTAVE in-plume conditions | | STRAP in-plume conditions | |
|---|---|---|---|---|---|---|
| | Median | $5^{th}$ – $95^{th}$ perct. | Median | $5^{th}$ – $95^{th}$ perct. | Median | $5^{th}$ – $95^{th}$ perct. |
| Rad. (W m$^{-2}$) | 540 | 19 - 1094 | 460 | 18 – 969 | 1078 | 1056 - 1107 |
| T (°C) | 15.9 | 12.5 – 18.1 | 13.9 | 10.8 – 19.7 | 10.5 | 10.2 – 11.4 |
| RH (%) | 65 | 29 – 89 | 68 | 16 – 87 | 66 | 52 – 74 |
| CS (s$^{-1}$) | $9.8\times10^{-4}$ | $5.3\times10^{-5} - 4.1\times10^{-3}$ | $5.3\times10^{-3}$ | $1.0\times10^{-3} - 1.0\times10^{-2}$ | $5.9\times10^{-3}$ | $2.6\times10^{-3} - 7.9\times10^{-3}$ |
| O$_3$ (ppb) | 37 | 24 – 47 | 39 | 31 – 52 | 58 | 51 – 63 |
| NO$_x$ (ppb) | 0.13 | BDL – 0.80 | 0.22 | BDL – 0.66 | 0.19 | 0.18 – 0.21 |
| SO$_2$ (ppb) | 0.73 | 0.43 - 1.11 | 11.75 | 3.03 – 75.54 | 10.36 | 2.38 – 22.82 |
| H$_2$SO$_4$ (cm$^{-3}$) | $4.2\times10^6$ | $3.6\times10^5 - 2.3\times10^7$ | $6.1\times10^7$ | $3.0\times10^6 - 3.4\times10^8$ | $1.6\times10^8$ | $9.4\times10^7 - 5.8\times10^8$ |

*BDL = Below Detection Limit

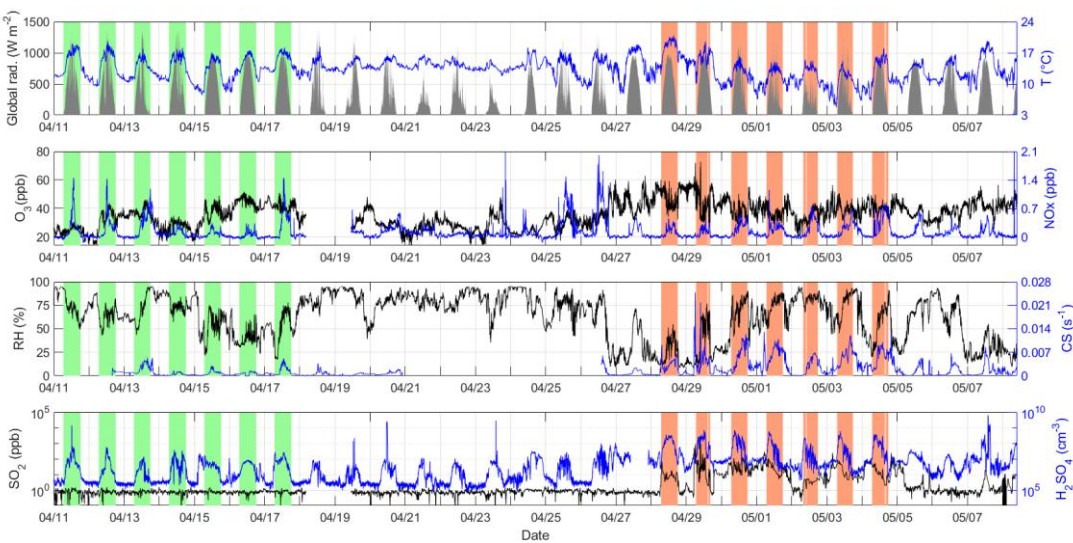

Fig. 1 Overview of the conditions during the OCTAVE campaign conducted in 2018. All time series are shown in Local Time (UTC +4) for easier interpretation of the diurnal cycles. Measurements performed in daytime regular conditions and used to derive the corresponding statistics reported in Table 1 are highlighted by the green shaded areas; similarly, data collected in daytime plume conditions are highlighted by the red shaded areas. Note that with the exception of radiation, which is represented by the grey area in the first panel, black line represents the variable on the left axis and blue line that on the right axis in all other panels. This information on line colour is not directly provided on the plot to preserve the readability of the figure.

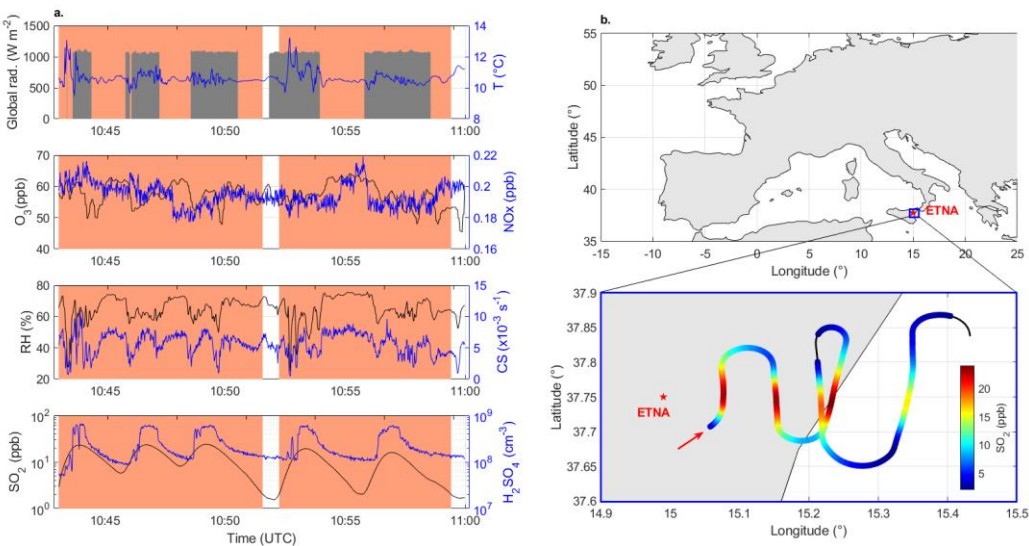

Fig. 2 Overview of the conditions during the first part of flight ETNA 13. a. Similarly to Fig. 1, measurements performed in plume conditions are highlighted by the red shaded areas. Note that with the exception of radiation, which is represented by the grey area in the first panel, black line represents the variable on the left axis and blue line that on the right axis in all other

panels. b. Map of the flight. Location of the plume is illustrated by the $SO_2$ levels reported on the coloured path, which direction is further indicated by the arrow. The red star indicates the position of the vent.

## 3. Proxy construction

The development of proxies for $[H_2SO_4]$ in volcanic plume conditions was performed with the exact same approach as in Mikkonen et al. (2011) and, more recently, Dada et al. (2020). Consequently, some of the aspects which were previously

described in details in the abovementioned studies are only briefly recalled here.

Just like in regular conditions, our current knowledge of $H_2SO_4$ production from reactions between $SO_2$ and OH radicals was first used as a basis for the construction of the proxies (Finlayson-Pitts and Pitts, 2000):

$$OH + SO_2 \rightarrow HSO_3 \tag{R1}$$

$$HSO_3 + O_2 \rightarrow SO_3 + HO_2 \tag{R2}$$

$$SO_3 + 2H_2O \rightarrow H_2SO_4 + H_2O \tag{R3}$$

Similarly, condensation on existing particles was also assumed to be the main sink of sulfuric acid, so the rate of change of $[H_2SO_4]$ could be expressed as follow:

$$d\,[H_2SO_4]/dt = k \times [OH] \times [SO_2] - [H_2SO_4] \times CS \tag{3}$$

where the temperature-dependant reaction rate k ($cm^3$ molecule$^{-1}$ s$^{-1}$) is given by DeMore et al. (1997) and Sander et al. (2002), and recalled in the Supplement.

[$H_2SO_4$] was then deduced from Eq. (3) assuming that the steady state between $H_2SO_4$ production and loss was also holding in volcanic plume conditions:

$$[H_2SO_4] = k \times [OH] \times [SO_2] \times CS^{-1} \tag{4}$$

Finally, as done earlier by Mikkonen et al. (2011) and more recently Dada et al. (2020), global radiation was introduced in Eq. (4) as an indicator of [OH] in order to further simplify the development of the proxies and allow as wide use as possible in future studies:

$$[H_2SO_4] = k' \times Global\ Radiation \times [SO_2] \times CS^{-1} \tag{5}$$

where k' corresponds to the multiplication of k by a factor (to be determined in the fitting procedure) which partly takes into account the use of global radiation instead of [OH]. The formulation of Eq. (5) was tested in proxy F1, which is similar to the proxy initially developed by Petäjä et al. (2009) and latter modified by Mikkonen et al. (2011) to take into account the temperature-dependant reaction rate k between $SO_2$ and OH. Considering the work by Mikkonen et al. (2011), we defined, in parallel, proxy A1, which was similar to F1 but with individual adjusted powers for the different variables, as such approach was reported to perform better to predict ambient sulfuric acid levels. In addition to F1 and A1, other proxies, all listed in Table 2, were tested to evaluate the importance of the sink term in determining [$H_2SO_4$]; note that "F" proxies include a single fitting parameter and fixed powers (1 or -1) for all variables, while "A" proxies allow adjusted fitting parameters for the different variables. Following the approach by Mikkonen et al. (2011), CS was first removed in proxies F2 and A2, and re-introduced in proxies F3 and A3 together with RH. The motivation for this last test was that CS may not always completely reflect the actual sink since the aerosol sample is systematically dried to below 40% RH at Maïdo. Applying a specific correction for the hygroscopic particle growth would have required a detailed characterisation of this process (*e.g.* as a function of air mass type, season) which has not been performed at Maïdo, and is likely not either available at a number of sites where the newly developed proxies could be used. Also, according to Mikkonen at al. (2011), such hygroscopicity correction might, at least in some environments, have only limited effect on the prediction of [$H_2SO_4$]. Therefore, similar to Mikkonen et al. (2011), inclusion of RH in the sink term was tested instead. In addition to A3, RH was also included in A4 but as an individual term. Based on earlier work by Mikkonen et al. (2011), the use of global radiation instead of [OH] was assumed to be accounted for in the fitting procedure for all these proxies, in specific by the mean of parameters K and a, respectively for the "F" and "A" proxies.

Finally, a last proxy, S1, was tested to take into account the loss of $H_2SO_4$ related to molecular cluster formation (Dada et al., 2020). This sink, written as a second-order function of [$H_2SO_4$] ($\beta \times [H_2SO_4]^2$), was reported to contribute up to ~35% of [$H_2SO_4$] prediction in Beijing, and we believe that it could also play an important role in the volcanic plume, where nucleation was previously reported to occur frequently (Rose et al., 2019). In order to account for the loss of $H_2SO_4$ due to cluster formation, Eq. (3) was modified based on the work by Dada et al. (2020):

$$d[H_2SO_4]/dt = \alpha \times k \times Global\ Radiation \times [SO_2] - [H_2SO_4] \times CS - \beta \times [H_2SO_4]^2 \tag{6}$$

where α is the coefficient associated to the $H_2SO_4$ production term (which accounts for the use of global radiation instead of [OH]) and β is the fitting parameter associated to the additional sink term.

Assuming again a steady state between $H_2SO_4$ production and loss, $[H_2SO_4]$ could be deduced from Eq. (6):

$$[H_2SO_4] = -\frac{CS}{2\beta} + \sqrt{\left(\frac{CS}{2\beta}\right)^2 + k\frac{\alpha}{\beta} \times Rad \times [SO_2]} \tag{7}$$

This last expression, hereafter referred to as proxy S1 (Table 2), is the same as the proxy derived from Eq. (4) in Dada et al. (2020), except that the reaction rate k was explicitly taken into account.

Additional variables, such as $O_3$ or $NO_x$ levels, could have also been introduced in the proxies, as done by Lu et al. (2019). However, since we did not observe a very specific behaviour of these species in the plume compared to regular conditions which could have motivated their inclusion, we rather chose to limit the number of variables to get as simple as possible

expressions for the proxies; only the inclusion of $[O_3]$, together with global radiation in the source term, was attempted in a sensitivity test (see Sect. 4.1). Similarly, the dependence of $H_2SO_4$ production term over absolute water concentration was left behind from the present work in order to avoid over-constraints which could prevent the use of the newly developed proxies in datasets collected in different volcanic plumes. More broadly, while Dada et al. (2020) explicitly aimed at understanding the different mechanisms of sulfuric acid formation and losses in different environments, detailed chemical investigation and/or

description of the formation pathways of $H_2SO_4$ and its precursors in a volcanic plume was behind the scope of the present work, which objective was, again, to obtain the simplest possible description of $[H_2SO_4]$ from a limited set of commonly measured variables.

The set of parameters leading to minimum sum of squared residuals was, for each proxy (*i.e.* K for F1-F3, a – f for A1-A4 and α - β for S1), determined iteratively using the *fminsearch* MATLAB function. The performance of the different proxies was

finally evaluated based on the correlation coefficient (R) of observed vs. predicted $[H_2SO_4]$ and corresponding sum of squared residuals (SSR), and, similarly to Mikkonen et al. (2011), by calculating the relative error (RE), which is defined as follows for a set of n observations:

$$RE = \frac{1}{n} \times \sum_{i=1}^{n} \frac{|[H_2SO_4]_{obs,i} - [H_2SO_4]_{proxy,i}|}{\overline{[H_2SO_4]_{obs}}} \tag{8}$$

where $\overline{[H_2SO_4]_{obs}}$ is the mean of observed $[H_2SO_4]$.

The data were in addition submitted to bootstrap resampling to evaluate the effect of a possible systematic error related to the measurement accuracy of $[H_2SO_4]$ and predictor variables on the fitting parameters and performance indicators (*i.e.* R, RE and SSR). The method is described in detail in Dada et al. (2020) and is only briefly recalled here. 10 000 bootstrap resamples were generated from the original dataset by randomly replacing an existing data point with another, and the resulting time series were further multiplied by a set of random factors to simulate the presence of independent systematic errors on the

different variables. For each variable, these factors (one per bootstrap sample, *i.e.* 10 000 in total) were drawn from a uniform distribution (in logarithmic scale) of possible biases in their respective uncertainty range. Specifically, uncertainties in the range between -50% and 100% were considered for measured $[H_2SO_4]$ (*i.e.* multiplying factors for $[H_2SO_4]$ in the bootstrap

resamples were between 0.5 and 2) following the work of Kürten et al. (2012). According to calibration data, we assumed an uncertainty of 15% in the measurement of $SO_2$ mixing ratio and, similar to Dada et al. (2020), we assumed 20% uncertainty in the CS evaluation. An uncertainty of 5% in the measurement of the remaining variables of interest (*i.e.* RH and global radiation) was finally accounted for based on manufacturer's specifications. For each function listed in Table 2, the fitting

procedure was first applied to the original dataset to obtain a set of reference parameters for deriving [$H_2SO_4$]. The variability of the fitting parameters and performance indicators was then evaluated for each proxy by repeating the same procedure on the bootstrap resamples.

Table 2 Proxy functions. F1-F3 are the proxies with powers fixed to -1 or 1 for all variables, as predicted by the theory, while

A1-A4 have individual adjusted powers for each variable. S1 includes the additional $H_2SO_4$ sink related to cluster formation. In each of the proxies, k corresponds to the temperature dependant reaction rate between $SO_2$ and OH. Fitting parameters K in F1-F3, a – f in A1-A4 and α - β in S1 were determined iteratively to minimise the sum of squared residuals associated to each proxy. The pre-factors a and K as well as parameter α are assumed to take into account the use of global radiation instead of [OH] in the different proxies.

| Proxy | Equation |
|---|---|
| F1 | $K \times k \times Rad \times [SO_2] \times CS^{-1}$ |
| F2 | $K \times k \times Rad \times [SO_2]$ |
| F3 | $K \times k \times Rad \times [SO_2] \times (CS \times RH)^{-1}$ |
| A1 | $a \times k \times Rad^b \times [SO_2]^c \times CS^d$ |
| A2 | $a \times k \times Rad^b \times [SO_2]^c$ |
| A3 | $a \times k \times Rad^b \times [SO_2]^c \times (CS \times RH)^e$ |
| A4 | $a \times k \times Rad^b \times [SO_2]^c \times CS^d \times RH^f$ |
| S1 | $-\dfrac{CS}{2\beta} + \sqrt{\left(\dfrac{CS}{2\beta}\right)^2 + k\dfrac{\alpha}{\beta} \times Rad \times [SO_2]}$ |

**4. Results**

**4.1 Derivation of the proxies using measurements performed in the volcanic eruption plume of the Piton de la Fournaise during OCTAVE**

As mentioned earlier, only the data collected in the volcanic eruption plume of the Piton de la Fournaise were effectively used to perform the fitting procedures, while the measurements obtained in the passive degassing plume of Etna were used in a

second step to further test the newly derived proxies in a different dataset. Such approach was motivated by the two main reasons which are listed below. We first believe that the variability of the key variables driving $H_2SO_4$ production was too

limited in the STRAP dataset (Fig. 2, Table 1) to retrieve a realistic picture of the role of these variables in predicting [$H_2SO_4$]. This was supported by the unphysical fitting parameters that we obtained when making an attempt to derive some of the proxies listed in Table 2 from this dataset. Also, while the questions related to the use of a proxy in different locations and/or conditions were raised in earlier studies, such questions obviously also apply to the volcanic plume, where the conditions are subject to

change very rapidly and also certainly differ depending on the type of plume (*e.g.* eruptive vs. passive degassing). We thus took the opportunity of using the data collected during STRAP to further evaluate how the proxies derived from OCTAVE could be used to predict [$H_2SO_4$] in a different volcanic plume with conditions similar to those encountered during OCTAVE. Before proceeding to the fitting, the few data points (8 in total, *i.e.* 0.5% of the measurements) corresponding to $SO_2 > 200$ ppb were removed from the OCTAVE dataset, as they were observed to significantly influence the fitting procedure, and due

to the limited number of measurements probably do not represent [$H_2SO_4$] under such large concentrations of $SO_2$ as a whole. The results of the fitting procedure are presented in Table 3. The following discussion focuses on the fitting parameters and performance indicators (*i.e.* R, RE and SSR) obtained for the original dataset, but Table 3 presents as well an estimate of their variability (25th and 75th percentiles) inferred from the bootstrap procedure introduced in the previous section. Note that in proxies A1-A4, the temperature – dependent reaction rate k was scaled by multiplying it with $10^{12}$ to get a more interpretable

estimated for a, as previously done by Mikkonen et al. (2011).

Based on the performance indicators reported in Table 3, better overall results were obtained with proxies A1-A4 (adjusted powers) in comparison to F1-F3 (fixed powers), as reflected by the higher correlation coefficients (0.70 – 0.80 vs. 0.51 – 0.71) and lower RE (0.43 – 0.57 vs. 0.49 – 0.69) and SSR (0.78 – $1.08 \times 10^{19}$ vs. 1.21 – $2.12 \times 10^{19}$ molecule² cm⁻⁶). This observation is in agreement with earlier results by Mikkonen et al. (2011), who found that individual adjusted powers for the different

variables improved the predictive ability of the proxies. In contrast, the lower prediction capability observed for F2 and A2 compared to F1 and A1, respectively, demonstrates that, unlike in the study of Mikkonen et al. (2011), the CS seemed to play an important role in regulating [$H_2SO_4$] in the volcanic plume of the Piton de la Fournaise. The need of taking the CS into account when predicting [$H_2SO_4$] was also recently highlighted by Lu et al. (2019) and Dada et al. (2020). However, we believe that while being less accurate than A1, A2 had a very simple expression which could still be used to get fair estimates of

[$H_2SO_4$] in absence of CS measurements in the volcanic plume. The reasonable results of A2 also indicate that [$SO_2$] and global radiation alone were capable of explaining a significant fraction of the variation of the sulfuric acid concentration in plume conditions, which was consistent with the relatively strong correlation observed between [$H_2SO_4$] and these two variables (0.45 and 0.68 for [$SO_2$] and global radiation respectively, Fig. S1). Including RH in F3 and A3, did not improve the results compared to F1 and A1. This observation either suggests that the inclusion of RH in the proxies was not adequate, or

that RH only had a minor effect on the sink term. Different inclusion of RH was tested in proxy A4 but lead to very similar results as A1, including identical prediction capability and similar powers for the variables they have in common. This observation thus suggests a minor role of RH on the sink regulating [$H_2SO_4$] production, in spite of the relatively high negative correlation observed between [$H_2SO_4$] and RH (-0.58, Fig. S1), and contrasts with the results obtained by Mikkonen et al. (2011), who noticed a better performance of the proxies when taking RH into account under regular conditions. Finally, the

comparison of proxies F1 and S1 showed an improvement of the proxy including the additional cluster sink term in comparison to the simple proxy, which nonetheless already had a good predictive ability, but the overall performance of S1 remained slightly lower compared to that of A1. Whether the overall better performance of A1 was due to the adjusted powers for the different variables, minor role or inadequate formulation of the cluster sink term in S1, remains however unknown. We in fact cannot exclude that the formulation of this additional sink term, which assumes that cluster formation mainly occurs via the stabilisation of $H_2SO_4$ by bases and was tested in regular conditions only (see Dada et al. 2020 and references therein), was not fully adapted to the specific conditions of the volcanic plume, because of the high concentration of sulphuric acid relative to the presence of bases.

The preceding analysis was in a first approach exclusively based on the performance indicators calculated for each proxy. However, these indicators mainly reflect the capability of the proxies to predict the highest $[H_2SO_4]$, as those logically have a very strong impact on the fitting procedure. While these high $H_2SO_4$ levels, characteristic of the volcanic plume conditions, were the main scope of this study, and, in turn, the target of the newly developed proxies, deeper analysis of the performance of the different proxies over the entire range of measured $[H_2SO_4]$ was performed in a second step. Figure 3 shows, for each proxy, the scatter plot between measured and predicted $[H_2SO_4]$. Correspondingly, Fig. 4 presents the cumulative sum of the squared residuals, which is used to illustrate the discrepancy between observed and predicted values throughout the range of measured $[H_2SO_4]$, and further point out that the overall performance of the proxies is determined by their capacity in predicting highest $[H_2SO_4]$. As shown in Fig. 4, sharp increases of the cumulative SSR were associated to few points, indicating strong discrepancies between the corresponding observed and predicted concentrations. Those were simultaneously observed for most of the proxies, but were on average more pronounced for proxies F1-F3. A possible explanation for that was the power of 1 (or -1) attributed to all the variables included in these proxies, which likely made the discrepancy caused by an extreme value of any of these variables stronger than for the proxies using lower powers (in terms of absolute value). This last observation thus highlights the fact that the interpretation of the SSR must be conducted carefully and crossed with other indicators, such as R or RE, as done in the first part of the analysis, or in the light of an additional view of the data, as done below.

Based on Figs. 3 and 4, three situations could be distinguished in the comparison of the proxies, corresponding to three subranges of measured $[H_2SO_4]$. Fig. 5 presents, in all three subranges and for each proxy, the median (as well as the 25th and 75th percentiles) of the ratio between predicted and measured $[H_2SO_4]$. As evidenced on Figs. 3 and 5, F1, F2 and, more importantly, F3 tended to underestimate $[H_2SO_4]$ in all subranges. In the case of F3, the systematic underestimation of the observations suggests that the sink resulting from the inclusion of RH together with the CS with the same weight as the source term was too high. However, besides F3, which had the worst results over the whole range of measured $[H_2SO_4]$, F1 and F2 gave the best predictions for $[H_2SO_4]$ below $2\times10^7$ cm$^{-3}$. In contrast, A1-A4 tended to overestimate these concentrations by factors between 3 and 6, with the largest discrepancies observed for $[H_2SO_4] < 1\times10^7$ cm$^{-3}$ (up to an average of one order of magnitude for A2 and, for instance up to 3 orders of magnitude higher cumulative sum of squared residuals for A2 compared to F1). The predictive ability of S1 was intermediate compared to that of the other proxies, with predicted concentrations on average slightly overestimated (factor of 1.8). The second $[H_2SO_4]$ subrange, between $2\times10^7$ and $2\times10^8$ cm$^{-3}$, was the one for

which all proxies (with the exception of F3) gave the most comparable results. The predictions of S1 were on average the closest to measured values, but were more dispersed compared to that of A1-A4, which also performed well over this range and retrieved [H$_2$SO$_4$] within a factor of 1.4 – 1.5 of observations. While remaining close to measured values, the predictions of F1 and F2 were on average slightly less accurate than for the lower concentrations. Over the last [H$_2$SO$_4$] subrange, which

was to a large extent influencing the overview retrieved by the performance indicators reported in Table 3, the worst results were obtained for F2 and F3. As shown on Fig. 5, the median ratio of predicted over measured [H$_2$SO$_4$] calculated for F1 was in contrast relatively close to unity (0.6) and to the median ratios calculated for A1, A3, A4 and S1. However, the predictions of F1 were more dispersed compared to that of the abovementioned proxies, as illustrated on Fig. 3 and also reflected on the cumulated SSR on Fig. 4, thus explaining its slightly lower performance depicted in Table 3. Finally, A2, which did not include

any sink term, showed the most pronounced deviations over A1-A4 in all three [H$_2$SO$_4$] subranges, further supporting the need for taking the CS into account when available. This deeper analysis thus confirmed the good performance of A1, A3 and A4 over the range of [H$_2$SO$_4$] which was the most relevant for the plume conditions. As already noticed, the advantage of using A3 and A4 was however limited, as they both required the knowledge of an additional variable (RH) which had almost no effect on the predictions. Besides A1, S1 also appeared as a good option. In fact, this proxy showed a better predictive ability

for [H$_2$SO$_4$] below ~$2\times10^8$ cm$^{-3}$ (up to 1 order of magnitude lower cumulative SSR), while performing also well at larger concentrations.

As a sensitivity test, an attempt to replace global radiation by the product Rad×[O$_3$] was made in all the proxies, to investigate if the explicit consideration of O$_3$, which photolysis is the main pathway for OH formation during daytime, would allow to further optimize the prediction of [H$_2$SO$_4$]. Corresponding results are reported in the Supplement (Table S1 and Figs. S3-S4),

but do not highlight any improvement in the performance of the proxies, which, with the exception of A2, all display worse performance indicators than when considering global radiation alone. Limited improvement in the predictive ability of the proxies was also noticed by Lu et al. (2019) when considering [O$_3$] with UVB in the urban atmosphere of Beijing, where the concurrent inclusion of [HONO] seemed in contrast to be more critical.

As a last step, we finally compared the predictions of proxies A1 and S1 with the results obtained with the proxy developed

by Mikkonen et al. (2011) (hereafter referred to as MIK), which expression is recalled below:

$$[H_2SO_4] = 8.21 \times 10^{-3} \times k \times Rad \times [SO_2]^{0.62} \times (CS \times RH)^{-0.13} \tag{9}$$

where k still corresponds to the temperature-dependant reaction rate between SO$_2$ and OH. This proxy was recently used by Rose et al. (2019) to predict [H$_2$SO$_4$] in the volcanic eruption plume of the Piton de la Fournaise in absence of direct measurements. While S1 had a slightly different structure, A1 was developed using the same approach as for MIK, and similar

features were observed in the two proxies, although they were initially dedicated to the description of different environments. As noticed earlier by Mikkonen et al. (2011), power b for global radiation was nearest to unity in A1 (0.81), thus indicating that, as in regular conditions, radiation was the main driving force for H$_2$SO$_4$ production in the volcanic plume, consistent with the high correlation already highlighted between [H$_2$SO$_4$] and global radiation (Fig. S1). Also, as previously observed by Mikkonen et al. (2011) and Lu et al. (2019), power c for [SO$_2$] was less than unity (0.51) and power d for CS was closer to

zero (-0.52) than assumed in Eq. (5) (and in turn proxy F1). In the conditions of the volcanic eruption plume of the Piton de la Fournaise, this discrepancy between observations and theory was likely explained, at least to a certain extent, by a connection between CS and $[SO_2]$, despite the absence of a statistically significant correlation between these two variables during OCTAVE (Fig. S1). The CS enhancement observed in plume conditions at Maïdo was however previously reported to result

mainly from secondary aerosol formation processes, which are expected to be tightly connected to $H_2SO_4$, and in turn $SO_2$, in such conditions (Rose et al., 2019).

Despite its similarity with A1, the behaviour of MIK was overall more comparable to that of S1 (R = 0.70, RE = 0.48 and SSR = $1.14\times10^{19}$ molecule$^2$ cm$^{-6}$), with improved performance in comparison to that of A1 for $[H_2SO_4] < \sim2\times10^8$ cm$^{-3}$, and slightly decreased prediction capability above this threshold (Fig. 6). In particular, MIK showed the best ability to reproduce $[H_2SO_4]$

in the range between $2\times10^7$ and $2\times10^8$ cm$^{-3}$, while being on average slightly less accurate compared to S1 at lower concentrations, with a median ratio of 2.0 between predicted and measured values. At larger concentrations, the predictions of MIK were slightly more underestimated compared to that of A1 and S1, but remained, on average, within a factor of 1.9 of measured values, against 1.4 and 1.5, for A1 and S1, respectively.

All in all, these results suggest that the newly developed proxies A1 and S1 slightly improved the predictions of the high

$[H_2SO_4]$ encountered in volcanic plumes, but also demonstrates the relatively good ability of the proxy developed earlier by Mikkonen et al. (2011) to also reproduce these concentrations. This last observation gives further confidence in the results recently obtained by Rose et al. (2019) by the mean of this last proxy for the investigation of NPF in the volcanic plume of the Piton de la Fournaise.

Table 3 Fit results. For each proxy, the correlation coefficient (Pearson) between observed $[H_2SO_4]$ and the predicted values (R), the relative error (RE) and the sum of squared residuals (SSR) are also reported. Numbers in brackets indicate the 25th and 75th percentiles of the corresponding fitting parameter or performance indicator derived from the bootstrap resamples. For simplicity, the results are reported separately for the three proxy families: a) F1-F3, which are the proxies with powers fixed to -1 or 1 for all variables, b) A1-A4, which have individual adjusted powers for each variable and c) S1, which includes the

additional $H_2SO_4$ sink related to cluster formation. Note that based on corresponding p-values, all correlations were found significant ($p < 0.05$).

a)

| Proxy | K | R | RE | SSR ($\times 10^{19}$ (molec. cm$^{-3}$)$^2$) |
|---|---|---|---|---|
| F1 | $1.25\times10^3$ (877-$1.76\times10^3$) | 0.71 (0.70-0.73) | 0.49 (0.48-0.50) | 1.21 (0.59-2.38) |
| F2 | $2.36\times10^5$ ($1.68$-$3.39)\times10^5$ | 0.57 (0.55-0.58) | 0.62 (0.60-0.63) | 1.88 (0.92-3.72) |
| F3 | $3.64\times10^4$ ($2.53$-$5.11) \times10^4$ | 0.51 (0.48-0.54) | 0.69 (0.67-0.70) | 2.12 (1.03-4.18) |

b)

| Proxy | a | b | c | d | e | F | R | RE | SSR (× 10^19 (molec. cm^-3)^2) |
|---|---|---|---|---|---|---|---|---|---|
| A1 | 5.30×10^-2 (2.67-9.54)×10^-2 | 0.81 (0.78-0.83) | 0.51 (0.49-0.53) | -0.52 (-0.54 – -0.50) | - | - | 0.79 (0.79-0.80) | 0.43 (0.42-0.44) | 0.78 (0.38-1.53) |
| A2 | 39.70 (23.41-64.57) | 0.60 (0.57-0.61) | 0.42 (0.40-0.43) | - | - | | 0.70 (0.70-0.71) | 0.57 (0.56-0.57) | 1.08 (0.52-2.12) |
| A3 | 3.30 (1.81-5.61) | 0.78 (0.75-0.80) | 0.45 (0.44-0.47) | - | -0.30 (-0.31– -0.29) | | 0.78 (0.77-0.79) | 0.45 (0.44-0.46) | 0.84 (0.41-1.65) |
| A4 | 7.16×10^-2 (3.19-12.01) ×10^-2 | 0.81 (0.78-0.83) | 0.50 (0.49-0.52) | -0.51 (-0.53 – -0.49) | - | -0.04 (-5.74 – -0.38) ×10^-2 | 0.80 (0.79-0.81) | 0.43 (0.42-0.44) | 0.78 (0.38-1.53) |

c)

| Proxy | α | β | R | RE | SSR (× 10^19 (molec. cm^-3)^2) |
|---|---|---|---|---|---|
| S1 | 3.29×10^3 (2.28-4.61) ×10^3 | 2.00×10^-11 (1.34-2.76) ×10^-11 | 0.75 (0.75-0.77) | 0.44 (0.43-0.45) | 0.93 (0.45-1.83) |

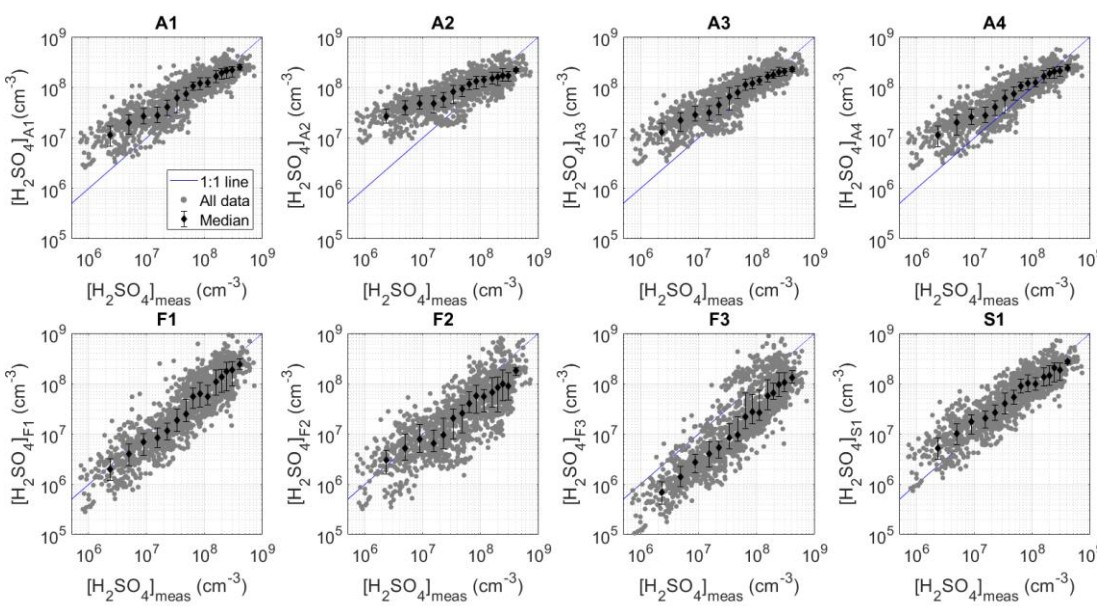

Fig. 3 Predictions retrieved by the different proxies as a function of measured [$H_2SO_4$]. The data points were in addition divided based on measured [$H_2SO_4$] into 15 bins with an equal number of data points. The medians (markers) and the 25th/75th percentiles (error bars) of [$H_2SO_4$] predicted in each bin are also shown.

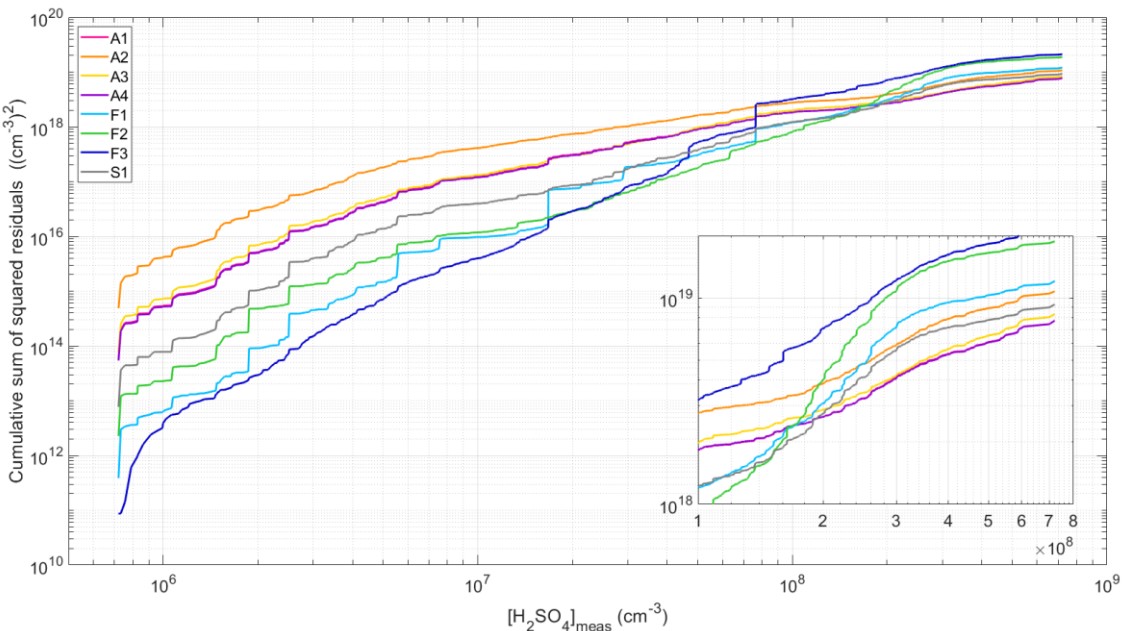

Fig. 4 Cumulative sum of squared residuals associated to the different proxies. The insert presents a zoom into the results obtained for $[H_2SO_4] > 1\times10^8$ cm$^{-3}$.

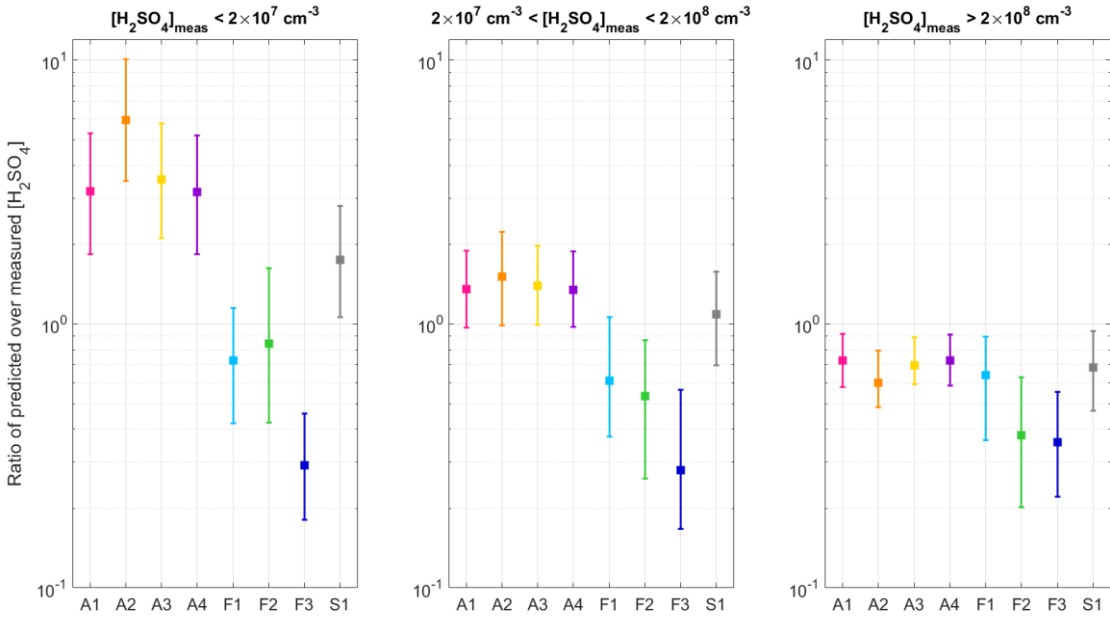

Fig. 5 Ratio between predicted and measured $[H_2SO_4]$ in the different $[H_2SO_4]$ subranges. For each proxy, the marker represents the median of the ratio, and lower and upper limit of the error bars indicate the 25[th] and 75[th] percentiles, respectively.

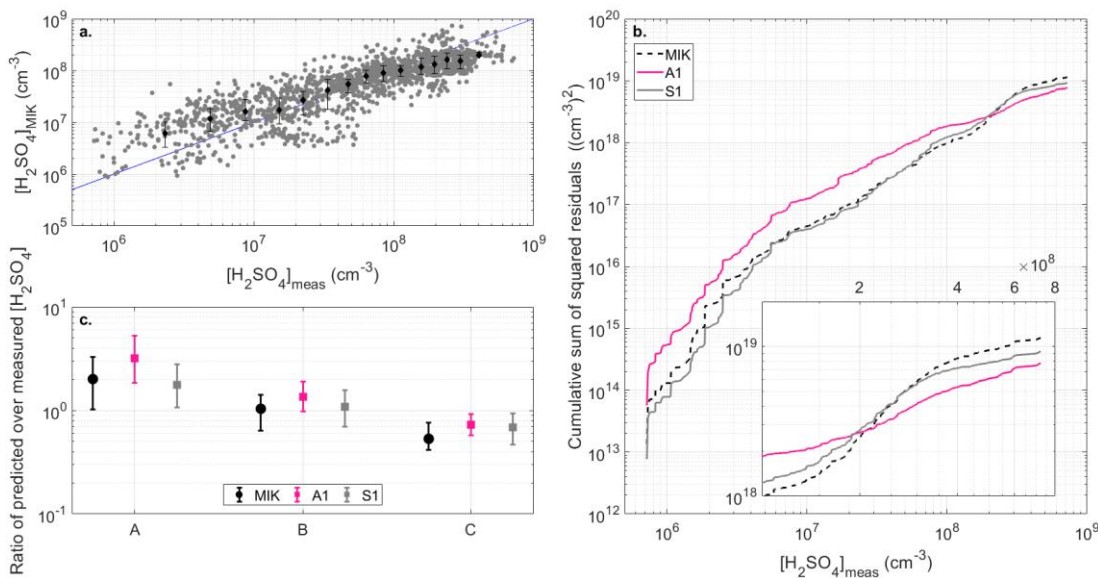

Fig. 6 Comparison between the results obtained with MIK and the predictions derived from A1 and S1. a. Predictions retrieved by MIK as a function of measured [$H_2SO_4$]. The data points were in addition divided based on measured [$H_2SO_4$] into 15 bins with an equal number of data points. The medians (markers) and the 25th/75th percentiles (error bars) of [$H_2SO_4$] predicted in each bin are also shown. b. Cumulative sum of squared residuals associated to MIK, A1 and S1. The insert presents a zoom into the results obtained for [$H_2SO_4$] > $1\times10^8$ cm$^{-3}$. c. Ratio between predicted and measured [$H_2SO_4$] in the different [$H_2SO_4$] subranges, with A corresponding to [$H_2SO_4$] < $2\times10^7$ cm$^{-3}$, B to [$H_2SO_4$] between $2\times10^7$ and $2\times10^8$ cm$^{-3}$ and C to [$H_2SO_4$] > $2\times10^8$ cm$^{-3}$. For each proxy, the marker represents the median of the ratio, and lower and upper limit of the error bars indicate the 25th and 75th percentiles, respectively.

### 4.2 Additional evaluation of the proxies using data collected in the passive degassing plume of Etna during STRAP

As mentioned above, the first part of flight ETNA 13, performed between 10:43 and 11:00 UTC on May 15th, was selected for further evaluation of the newly derived proxies. As shown in Section 2.3, the conditions encountered during this part of the flight were, for most of the investigated variables, comparable to the average conditions observed in the volcanic plume of the Piton de la Fournaise. However, unlike during OCTAVE, most of these variables only showed very limited variability. In specific, [$H_2SO_4$] measured in the passive degassing plume of Etna were mainly above $2\times10^8$ cm$^{-3}$, *i.e.* among the highest concentrations measured during OCTAVE and in the [$H_2SO_4$] range where contrasting performances were previously observed for the different proxies.

As shown on Figs. 7 and 8, proxies F1-F3 were found to systematically underestimate [$H_2SO_4$], and were in turn associated to higher RE and SSR compared to the other proxies (Table 4). Highest correlation coefficient was however obtained for F2 (R = 0.80), closely followed by A2 (R = 0.77), which both did not include any sink term. As illustrated on Fig. 7, this strong

correlation between calculated and observed [$H_2SO_4$] was likely explained by the fact that the predictions of F2 and A2 were overall less dispersed compared to that of A1, A3 and A4, and, more importantly, F1 and F3, for which the dependence of predicted [$H_2SO_4$] over CS was the strongest (power -1), and resulted in the lowest correlation coefficients. More than the question related to the inclusion of the CS itself, these observations raised the question of the relevance of the fitting parameters

derived from OCTAVE for further application in the STRAP dataset. In fact, as mentioned in Section 2.3, the CSs were not calculated over the same size range in the two datasets (10-600 nm during OCTAVE and 90-3000 nm during STRAP), and the values calculated for STRAP were likely a lower limit of the actual sink. Also, in spite of the significant correlation observed between the CS and [$SO_2$] (Fig. S2), the origin of the CS during STRAP might have been less connected to the plume than during OCTAVE, and instead more related to the intrusion of more polluted boundary layer air masses at higher altitude,

as suggested by the strong link between the CS and RH (Fig. S2). Consequently, the hypothesis of a common origin for the CS and [$SO_2$] explaining the "balance" observed between the powers of these two variables in A1, A3 and A4 may have not held during STRAP. Regarding the last investigated proxy, S1, somewhat intermediate performance was observed, as previously noticed in the OCTAVE dataset. Indeed, the predictions of S1 were overall closer to measurements compared to those of F1-F3, while being on average not as good as those of A1-A4, as reflected by the corresponding RE and SSR (Table

4). On the other hand, the correlation coefficient obtained for S1 (R = 0.60) was in between the high values obtained for F2 and A2 and those of the remaining proxies.

Besides the CS, which had an obvious effect on the predictions, it was concurrently seen that improved correlations between predicted and observed [$H_2SO_4$] were obtained when the dependence over [$SO_2$] was the highest in the proxies, as illustrated in particular by the stronger correlation obtained for F2 compared to A2. This result was consistent with $SO_2$ being the main

driver of [$H_2SO_4$] variability in the context of STRAP, as suggested by the strong correlation observed between [$SO_2$] and [$H_2SO_4$] (Fig. S2), also reflected on the time series presented on Fig. 2.a. Note that for [$SO_2$], each single reported observation was actually the result of the past 20 seconds of measurement, thus explaining the smoother apparent variations compared to that of [$H_2SO_4$]. Based on these last observations, it was thus highly probable that in A1, A3 and A4, in addition to the already mentioned possible issues related to the inclusion of the CS in the proxies, the dependence over [$SO_2$] was too weak to

reproduce properly the high [$H_2SO_4$] observed in the context of STRAP. As illustrated on Fig. 8.a, the predictive ability of A1-A4 was in particular decreased for [$H_2SO_4$] > $3\times10^8$ cm$^{-3}$, thus suggesting that, among other possible factors, the dependence of [$H_2SO_4$] over [$SO_2$] was not well represented over this range of concentrations. Note that a possible effect of RH and global radiation could not be excluded. However, we believe that inadequate fitting parameters for these two variables would have equally affected all predicted concentrations due to their limited variability. This was for instance illustrated for F3, in which

the inclusion of RH lead to a systematic underestimation of the concentrations and relatively constant difference with the predictions of F1 over the whole range of measured [$H_2SO_4$].

In the case of F1-F3, which had a stronger dependence over [$SO_2$], the average ratio between predicted and observed [$H_2SO_4$] was in contrast more comparable below and above the identified threshold concentration. This observation suggested that the predictions of these proxies, while being obviously affected by the inclusion of CS and/or RH, could primarily be

systematically underestimated due to inadequate values of the pre-factor K. This hypothesis was tested for F2, which had the simplest formulation and displayed the closest ratios (0.31 and 0.23) on both sides of the threshold concentration. For that purpose, the fitting procedure was repeated for F2 using the STRAP data to derive F2', and the adjusted pre-factor K' $(9.27\times10^5)$ significantly improved the results compared to that derived from OCTAVE, as illustrated on Fig. 9 (R = 0.80, RE

= 0.26 and SSR = $0.64\times10^{19}$ molecule² cm⁻⁶). Note that the evaluation of a possible systematic error related to the measurement accuracy of [$H_2SO_4$] and predictor variables on the fitting parameters and performance indicators was left behind this last test, which purpose was simply to get an estimate of the improvement in proxy performance related to the derivation of location specific coefficients. As a last analysis, the predictions of F2' were finally compared to that of MIK (Fig. 9), which were on average less accurate and logically more comparable to that of A1, A3 and A4, due to their very similar structures. The

correlation between predicted and observed [$H_2SO_4$] was higher for MIK (R = 0.72) compared to A1, A3 and A4, as a likely result of the lower power of the CS in MIK (-0.13 in MIK against -0.52, - 0.30 and -0.51 in A1, A3 and 14, respectively), which otherwise displayed similar RE and SSR (0.57 and 2.98 $\times10^{19}$ molecule² cm⁻⁶, respectively).

All in all, these last results demonstrate the ability of the proxies derived from OCTAVE to fairly predict [$H_2SO_4$] in the plume of Etna, but they concurrently highlight a limited improvement of the predictions compared to MIK. Together with the

15 improved performance of F2' over F2, this observation illustrates that in volcanic plumes like in other environments, location specific coefficients do logically increase the ability of the proxies to reproduce measured concentrations.

Table 4 Ability of the newly developed proxies to predict [$H_2SO_4$] measured during flight ETNA 13. For each proxy, R is the correlation coefficient (Pearson) between predicted and observed [$H_2SO_4$], RE is the relative error and SSR is the sum of squared residuals. Note that based on corresponding p-values, all correlations were found significant (p < 0.05).

| Proxy | R | RE | SSR ($\times 10^{19}$ (molec. cm⁻³)²) |
|---|---|---|---|
| F1 | 0.23 | 0.74 | 4.69 |
| F2 | 0.80 | 0.75 | 4.55 |
| F3 | 0.11 | 0.87 | 6.09 |
| A1 | 0.35 | 0.54 | 2.83 |
| A2 | 0.77 | 0.51 | 2.59 |
| A3 | 0.48 | 0.53 | 2.72 |
| A4 | 0.35 | 0.54 | 2.83 |
| S1 | 0.60 | 0.60 | 3.28 |

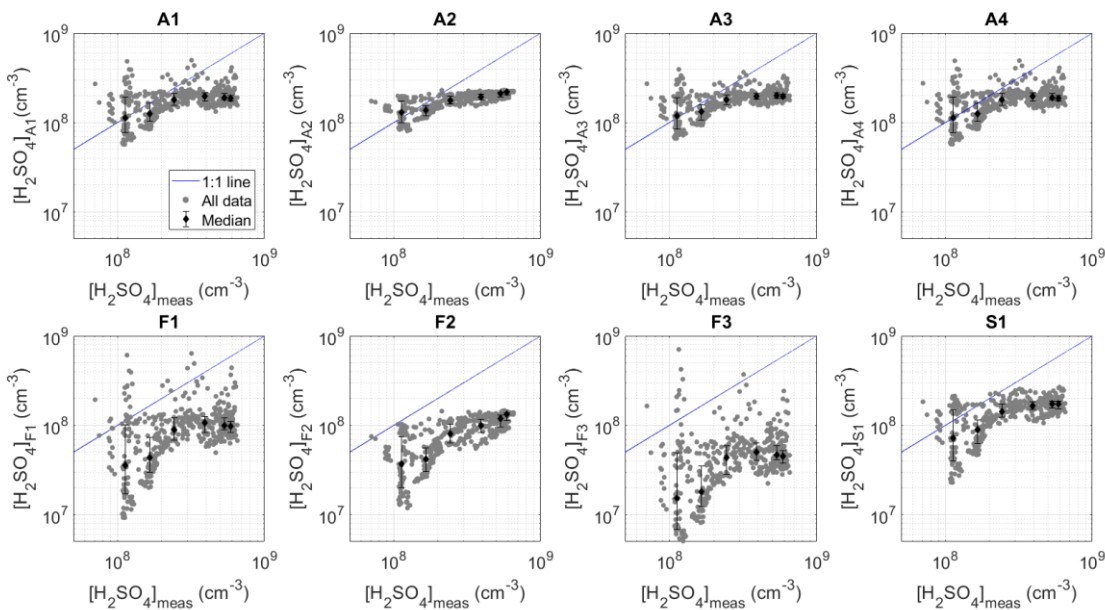

Fig. 7 Predictions retrieved by the different proxies as a function of [H$_2$SO$_4$] measured during STRAP. The data points were in addition divided based on measured [H$_2$SO$_4$] into 6 bins with an equal number of data points. The medians (markers) and the 25$^{th}$/75$^{th}$ percentiles (error bars) of [H$_2$SO$_4$] predicted in each bin are also shown.

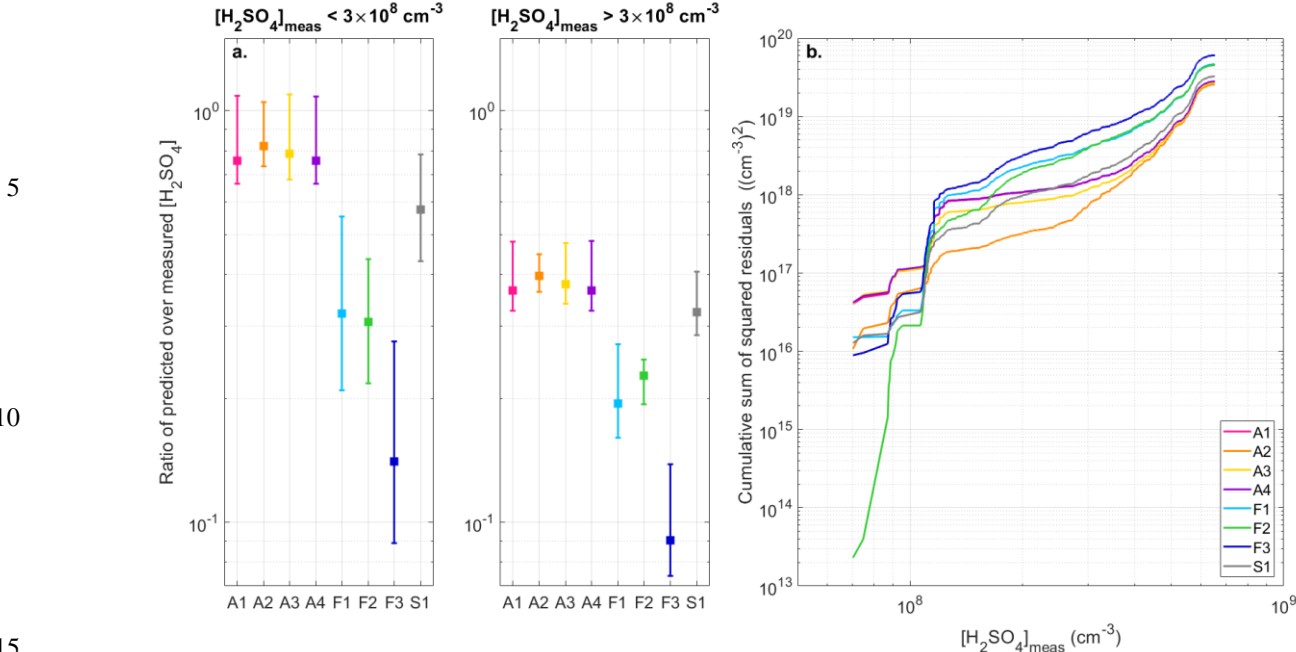

Fig. 8 a. Ratio between predicted and measured [H₂SO₄], separately for the concentrations below and above $3\times10^8$ cm⁻³. For each proxy, the marker represents the median of the ratio, and lower and upper limit of the error bars indicate the 25th and 75th percentiles, respectively. b. Cumulative sum of squared residuals associated to the different proxies.

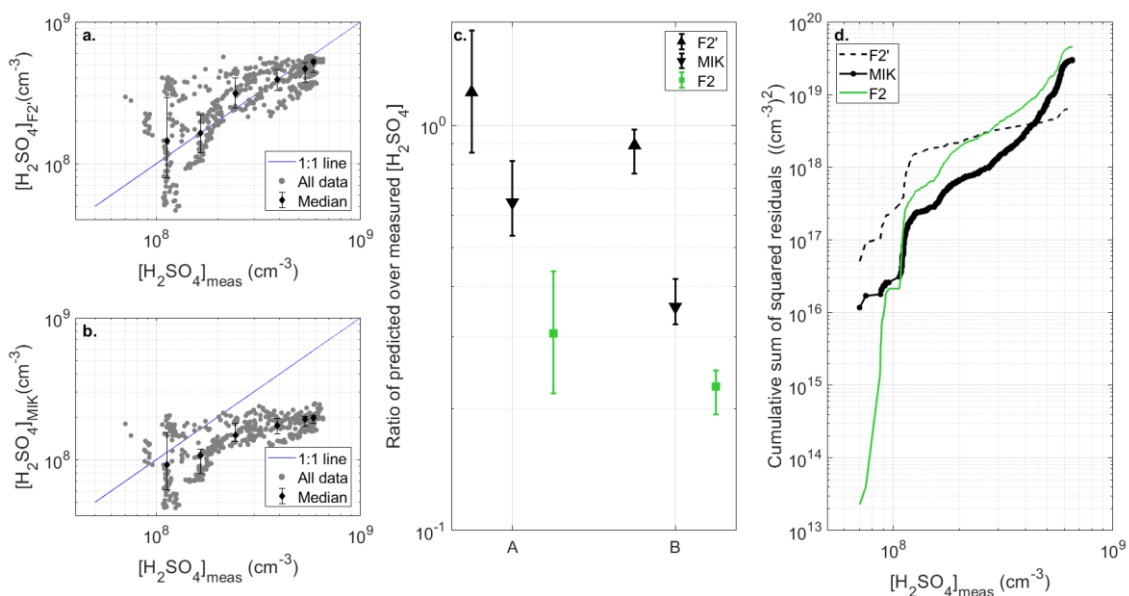

Fig. 9 a. Predictions retrieved by the proxy F2' as a function of [H₂SO₄] measured during STRAP. b. Same as a. for MIK. In a. and b., the data points were in addition divided based on measured [H₂SO₄] into 6 bins with an equal number of data points.

The medians (markers) and the $25^{th}/75^{th}$ percentiles (error bars) of $[H_2SO_4]$ predicted in each bin are also shown. c. Ratio between predicted and measured $[H_2SO_4]$, separately for the concentrations below (corresponding to A) and above (corresponding to B) $3\times10^8$ cm$^{-3}$. The results obtained with F2 are also reported for direct comparison. For each proxy, the marker represents the median of the ratio, and lower and upper limit of the error bars indicate the $25^{th}$ and $75^{th}$ percentiles, respectively. d. Cumulative sum of squared residuals associated to the different proxies.

## 5. Summary and conclusions

Measurements recently performed in the volcanic plumes of the Piton de la Fournaise and Etna have featured sulphuric acid concentrations approaching $10^9$ cm$^{-3}$. These concentrations are, to our knowledge, the highest ever recorded in the atmosphere, and have in turn motivated the present work, which main objectives were to 1) investigate the performance, in these conditions, of the proxies available in the literature for the prediction of $[H_2SO_4]$ and 2) develop proxies adapted to these very specific conditions.

Data collected at Maïdo (La Réunion Island) in the plume of the Piton de la Fournaise, during the OCTAVE campaign which took place in March-May 2018, were used in a first approach. We overall followed the same approach as Mikkonen et al. (2011) to develop proxies able to predict daytime $[H_2SO_4]$ (global radiation > 10 W m$^{-2}$), assuming that oxidation of $SO_2$ by the hydroxyl radical OH was the only source of $H_2SO_4$ and that [OH] could be replaced by global radiation for simplicity. The condensation sink (CS) was in a first approach considered as the only sink contributing to the balance of $[H_2SO_4]$ to derive seven proxies based on the knowledge of $SO_2$ mixing ratios, global radiation and CS. The inclusion of RH in the sink term was in addition tested in several formulations. In three of the seven proxies (F1-F3), power 1 or -1 was attributed to all included variables, thus giving them equal weight in the prediction of $[H_2SO_4]$, while adjusted powers were allowed for the different variables in the remaining four proxies (A1-A4). In the light of the recent work of Dada et al. (2020), a last expression was tested, S1, which includes an additional sink term related to molecular cluster formation.

Proxies A1-A4 were overall found to perform better compared to F1-F3 in the plume of the Piton de la Fournaise, with, in specific, improved predictive ability for $[H_2SO_4] > 2\times10^8$ cm$^{-3}$. The CS was observed to play an important role in regulating $[H_2SO_4]$, but proxy A2, which did not include the CS contribution, was however able to retrieve fair estimations of $[H_2SO_4]$, thus indicating that such simple expression could be used in absence of aerosol data. The reasonable results obtained with A2 demonstrated at the same time that, as also observed in regular conditions, $SO_2$ and global radiation alone were capable of explaining a significant fraction of the variation of the sulfuric acid concentration, consistent with the strong connection found between $[H_2SO_4]$ and these key variables. In contrast, the inclusion of RH, either with the CS or with a separate power, did not improve the performance of proxies A3 and A4, respectively, compared to A1. This observation suggested a limited effect of RH on the sink regulating $H_2SO_4$ production, in spite of the relatively high negative correlation observed between these two variables. More importantly, power -1 attributed to RH in proxy F3 lead to a systematic underestimation of $[H_2SO_4]$. Finally, proxy S1, which had a somewhat different structure compared to the other proxies and included the additional cluster sink

term, also showed a very good predictive ability, close to that of A1 for $[H_2SO_4] > 2\times10^8$ cm$^{-3}$, but on average higher at smaller concentrations. The capacity of the proxy initially developed by Mikkonen et al. (2011) for the prediction of $[H_2SO_4]$ in regular conditions to also reproduce $[H_2SO_4]$ in the plume of the Piton de la Fournaise was finally evaluated against A1 and S1. Despite being on average lower compared to that of the newly developed proxies, the predictive ability of the proxy from that work

appeared to be surprisingly good, often close to that of S1.

In a second step, the newly developed proxies were tested in a different dataset, collected on May 15$^{th}$ in the passive degassing plume of Etna, during the first part of flight ETNA 13 (10:43 and 11:00 UTC, *i.e.* LT -2h) performed in the frame of the STRAP campaign. This specific time period was selected as several latitudinal plume transects were performed at constant altitude (~ 2900 m) at distances between ~ 7 and 39 km from the vent, resulting in very clear variations of $[H_2SO_4]$. Also,

while only showing little variability, the conditions encountered during this part of the flight were, for most of the investigated variables, similar to the average conditions observed in the plume of the Piton de la Fournaise. In the case of Etna, increased correlations between observed and predicted sulphuric acid concentrations were obtained when the dependence of predicted $[H_2SO_4]$ over CS was the lowest, and when the dependence over $[SO_2]$ was concurrently the highest. In fact, the presence of CS in the proxies resulted in scattered predictions, and underestimated power for $[SO_2]$ was observed to affect their predictive

ability, in particular for $[H_2SO_4] > 3\times10^8$ cm$^{-3}$. As also seen for the volcanic eruption plume of the Piton de la Fournaise, albeit to a lesser extent, the proxy by Mikkonen et al. (2011) was able to provide reasonable predictions of $[H_2SO_4]$ in the STRAP dataset, very close to that of A1, A3 and A4. The best predictions were finally retrieved by the simple formulation of F2 (which did not consider CS and had the maximum possible dependence over $[SO_2]$), with a pre factor adapted to the STRAP data.

All in all, our results illustrate the fairly good capacity of the proxy developed by Mikkonen et al. (2011) to also describe

$[H_2SO_4]$ in volcanic plume conditions, but highlight at the same time the benefit of the newly developed proxies dedicated to these specific conditions for the prediction of the highest concentrations ($[H_2SO_4] > 2\text{-}3\times10^8$ cm$^{-3}$). Also, the contrasting predictive ability of the new proxies in the two different datasets, OCTAVE and STRAP, indicates that in volcanic plumes like in other environments, the relevance of a proxy can be affected by changes in environmental conditions, including in this particular case the type of plume (*e.g.* passive vs. eruptive), in connection with the variable nature of the volcanic eruptions.

Like in other environments, location specific coefficients thus logically improve the predictive ability of the proxies.

**Data availability**: DMPS data as well as NOx and O$_3$ mixing ratios from Maïdo are available from the EBAS data center. Meteorological parameters, SO$_2$ mixing ratios as well as CI-APi-TOF data are available upon request. Measurements performed in the frame of STRAP are available on the STRAP website (http://osur.univ-reunion.fr/recherche/strap/database/).

**Author contributions**: JB, AC and MPR contributed to the coordination of the OCTAVE 2018 campaign and MK provided additional financial support for travel cost during the campaign. KS and PT contributed to the coordination of the STRAP project. CR, MPR, SI, AC, XCH, JL, YJT, DW, JMM and PT performed the measurements during the OCTAVE campaign,

and KS, AC, RD and JD performed measurements during the STRAP campaign. CR, MPR, SI, JD, CY, JBN, AC, RD and KS analysed the data. CR and MPR wrote the paper. All co-authors contributed to reviewing the manuscript.

**Acknowledgements**: The OCTAVE 2018 campaign was performed in the frame of the project OCTAVE of the "Belgian Research Action through Interdisciplinary Networks" 5 (BRAIN-be) research program (2017-2021) through the Belgian Science Policy Office (BELSPO) under the contract number BR/175/A2/OCTAVE. We would like to thank UMS3365 of OSU-Réunion for its support to the deployment of the instruments. The French programme SNO-CLAP is also acknowledged for supporting continuous aerosol measurements at the Maïdo observatory.

Airborne data collected during STRAP were obtained using the aircraft managed by SAFIRE, the French facility for airborne research, an infrastructure of the French National Center for Scientific Research (CNRS), Météo-France and the French National Center for Space Studies (CNES).

**Financial support**: Support was received from the European Union's Horizon 2020 research and innovation programme (ACTRIS TNA, grant agreement No 654109) to organize part of the measurements conducted during the OCTAVE 2018 campaign. MPR also appreciates funding from the Academy of Finland (project numbers: 299574, 326948 and 331207).

The STRAP project was funded by the Agence Nationale de la Recherche (ANR-14-CE03-0004-04). Part of the flight hours during STRAP were funded by the ClerVolc project-Programme 1 "Detection and characterization of volcanic plumes and ash clouds" funded by the French government "Laboratory of Excellence" initiative, ClerVolc contribution number 311.

DW is finally thankful for the support received from the Austrian Science Fund (FWF, Project J3951-N36).

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
