# Peer review of "Investigation of several proxies to estimate sulfuric acid concentration in volcanic plume conditions"

_Atmospheric Chemistry and Physics, 2020_

## Referee Comment (RC1) · Anonymous Referee #3 · 9 Oct 2020

General Comments: This manuscript presents a comparison of H2SO4 proxies used to estimate sulfuric acid in a volcanic plume. While there are various proxies in the literature to estimate H2SO4 (a difficult species to measure) their use in environments with large SO2 and thus H2SO4 concentrations has not been explored. To this end, using previously published approaches, the authors developed 8 different proxies for estimating H2SO4 concentrations using various parameterizations SO2 concentrations, solar radiation, condensation sink and relative humidity. These proxies were then compared against H2SO4 measurements of the volcanic plume from the Maïdo observatory located on La Réunion Island. Results showed that the proxies where the weights of the individual parameters were adjusted gave the best agreement compared to mea-

surements. A proxy using an additional sink term related to cluster formation gave good agreement over the whole range of measured H2SO4 concentrations. These proxies were also compared to airborne measurements of H2SO4 concentrations of the volcanic plume from Mt. Etna. Here the best proxy-measurement agreement was observed when the dependence of predicted H2SO4 concentration to CS was the lowest, and when the dependence on SO2 concentration was simultaneously the highest. Comparison of an existing literature proxy with the two data sets also gave reasonable agreement in these high H2SO4 concentration environments, yet not to the level of the newly developed proxies.

In all I found this paper to be well thought out and written. One aspect I found to be missing was a discussion of the uncertainties in both the proxies and the measurements. H2SO4 measurements in general have associated uncertainties on the order of +/- 35-45%, however in this work none were reported. The parameters used in the proxy calculations also have associated uncertainties resulting in an overall uncertainty in the estimated H2SO4 concentration. These uncertainties (both the measurements and calculations) need to be discussed in terms of the comparisons.

Also at such high H2SO4 concentrations (approaching 10e9 molecule cm-3), the reagent ion will be depleted in the NO3 CIMS measurements. The stated concentration calculation assumes pseudo first order kinetics which most likely are no longer applicable under these conditions. Were calibrations performed using these large H2SO4 concentrations to prove the validity of the calculation?

Another aspect was the discussion of the airborne measurements. The best agreement between proxy and measurements was observed when no CS term was included (not very realistic). It was also stated that there were gaps in radiation data caused by improper measurements during turns, when the aircraft itself affected the amount of radiation reaching the sensor. Was upwelling radiation considered in these calculations? Assuming these measurements were made inside the plume with particles present, the nadir or reflected radiation could approach that of the zenith or incoming.

An increase in the H2SO4 production term could balance the inclusion of the CS term.

I would find a statement about the estimated lifetime of gas phase H2SO4 under these high H2SO4 conditions useful too.

Below are some specific comments and suggested grammatical changes. One large one to mention here is the omission of the proxy equation (Eq. 11) from Mikkonen 2011. It is useful and relevant to this work and not proper to ask the reader to look up.

I feel that if these comments and suggestions are addressed, this work would be a fine contribution to ACP.

Specific Comments:

All Fig. X should be Figure X in text

Consider replacing "regular conditions" with "low SO2 conditions"

Page 6, line 19: with had a -> which had a

Page 6, line 27: in and off-plume -> in and out of plume

Lower median radiation during STRAP then OCTAVE due to diurnal observations during OCTAVE. Why not compare same time of day?

Page 7, line 1: in and off-plume -> in and out of plume (you use outside plume in line 3)

Figure 1 - Why not use black for radiation similar to other plots. Yellow for radiation is hard to see. Consider changing the color.

Page 10, lines 1-3: Equation is busy and unnecessary

Page 10, line 10: define k'

Page 10, Line 11: we defined in parallel proxy A1 -> we defined a parallel proxy A1

Page 10, line 18: CS was first removed in proxies F2 and A2, and it was reintroduced... -> the CS was first removed in proxies F2 and A2, and re-introduced...

Page 10, line 22: respectively for "F" and "A" proxies. -> respectively for the "F" and "A" proxies.

Page 10, line 24: contribute up to ~35% to -> contribute up to ~35% of

Page 10, lines 29-30: definitions of alpha and beta need to come earlier.

Page 11, line 13: which is defined as follow for -> which is defined as follows for

Page 11 Table 2: parameters need to be defined, especially k. Is K x k = k' in equation 6? As general rule the reader should not have to read the text to understand a figure or table.

Page 12, line 12: in a different volcanic plume, in which the conditions however overall resembled the average conditions encountered during OCTAVE. -> in a different volcanic plume with conditions similar to those encountered during OCTAVE.

Page 12, line 13: and we believe that their behaviour was not well caught due to their limited number. -> and due to the limited number of measurements probably do not represent H2SO4 concentrations under such large concentrations of SO2 as a whole.

Page 12, line 30: measurement -> measurements

Page 13, line 1: make them better -> improve results

Page 13, line 6: in regular conditions, who noticed a better performance of the proxies when taking RH into account. -> who noticed a better performance of the proxies when taking RH into account under regular conditions.

Page 13, line 22: point up that overall -> point out that the overall

Page 13, line 23: shown on -> shown in

Page 14, line 23: Eq. (11). At first look I thought this referred to Eq. (11) in the present work, which doesn't exist. I'm assuming this refers to Eq. (11) in Mikkonen et

al. (2011). The equation needs to be added here so the reader doesn't have to go look it up!

Page 19, line 5: CS -> CSs

Page 23, line 15: The condensation sink (CS) was in a first approach considered as the only sink contributing to the balance of [H2SO4] to derive seven proxies based on the knowledge of SO2 mixing ratios, global radiation, CS and RH, which inclusion in the sink term was tested in several formulations. -> The condensation sink (CS) was in a first approach considered as the only sink contributing to the balance of [H2SO4] to derive seven proxies based on the knowledge of SO2 mixing ratios, global radiation, and CS. RH, included in the sink term was tested in several formulations.

Page 23, line 28: In contrast, the inclusion of RH,. . . This sentence is long and runs on. Needs to be rewritten and split up to convey conclusions.

Page 24, line 1: and on average higher at – but was on average higher at

Page 24, line 4: the literature -> that work

---

## Referee Comment (RC2) · Anonymous Referee #2 · 29 Nov 2020

Review of Rose et al. Sulfuric acid proxies. This paper tries out various parameterizations for sulfuric acid in the high SO2 conditions of a volcanic plume. Motivation for this work seems thinly presented, e.g. is an aim to provide a simple equation for GCMs? One set of field data was simulated somewhat reasonably but the other was not. Is that a sign that these proxies are limited in their application? A third finding (p.23 line 11) could be the limited scope of SA proxies and thus a chemical model (e.g. Boy 2005) will be needed in lieu of widespread measurements of SA.

Some detailed comments

p1,line20 'all variables equal weight'? Do F1, F2, and F3 represent just cases of different fitted forms? 'adjusted powers' is this varying weights in the fits or also the fitted forms? Succint description of the main analyis in the abstract should not mislead a potential reader...

p1.line25 But it seems S1 did not do well for the second case ?

p2.lines21-24. This is a mangled sentence that needs to be parsed out or eliminated. The next sentence begins a new thought and deserves a new paragraph.

p3.line16. So a main motivation was to duplicate the Lu et al. study? Needs rewording.

p3.line22. Points out a flaw in the proxy idea in general: missing information is important such as NO, NO2, CH2O, O3, radiative environment, aerosol hygroscopicity ?

p3.line25. This is an unusual statement: they used the proxies but did not test them ? Do you mean compared to measured H2SO4? Did not Rose 2019 have measurements?

p4.line15. 'compared to' -> 'than'. Nonetheless, why care about the previous one?

p4.line16. One has O3 and NOx too... seems that a Boy 2005 model simulation could be done.

p4.line24. made a non-negligible contribution to what ? Read on... Is there ion-molecule clustering going on or are these ionization of neutral clusters? Should 195u be multiplied by 2 and 293u by 3 to get the total SA being produced? How does one separate ionization of H2SO4 dimer and trimer from the clustering of HSO4- with H2SO4? Still no reference for this equation. Or for the AI instrument, next page. Was the calibrations carried out to test the limits of these equations?

p5.line30-32. These two sentences add nothing to the paper, instead they distract. Instead here is where some discussion of the uncertainties in the H2SO4 measurements and whether the particulars of the deployment(s) have affected them. Did exp. conditions perhaps introduce more factors (e.g. ambient pressure dependencies of the C factors) ?

p7.line3 Boy 2005 showed that NO +HO2 was a major source. In fact, why not run that model in the plume for comparison?

p10.line8. This seems to be not only wrong chemically but misguided. O3 times radiation (hv) is the correct parameter and you should use it as such. Let potential future users input the ozone for their situation. Also, when water is low such as cold temperatures at altitude, quenching of O(1D) by air molecules would compete. Is SO2 in the plume so high that it is optically thick? Would the UVB be affected more than global radiation (this reveals that 'global radiation' and its measurement should be explained in depth in this context)? Again, Boy 2005 showed that NO + HO2 was the major direct source for OH. All these questions could be addressed by some sort of OH box model run for plume conditions.

p10.line10. kprime is what? The parameter K must have some physical meaning.

Also, CS concerns here. On the face of it, the CS used is not correct and one should apply correction factors as the aerosol is probably hygroscopic (loads of SO2 here.) Furthermore: below 40%? The level of drying is not even known? Uncertainties pile up. A nominally hygroscopic aerosol should be assumed and the CS should corrected as best as possible, perhaps using half the nom.hyg.aer. correction.

p10.line25 That is huge,35%. It would be a dominant contribution to the signal at 195u.

p10.line30 Give some insight on this term. It represents a net flux to the dimer. That this is a constant for a whole measurement campaign is not realistic. Agreement with any estimated dimer abundance / loss rate should be discussed.

p11.line9. What ? Essentially saying: We don't want to know too much. Also arguing that proxies are not meant to be all that useful? Instead, put as much as you know into it. Then knowledge develops, limitations in it become known.

These last few comments underscore the many levers of uncertainty here. The final section has too-much-summary of the various proxies and there is not enough presentation of a firm conclusion regarding the limits of their chosen proxy (let's not have 8 or 9 to choose from) in the light of the uncertainties.

---

## Author Comment (AC1) · 27 Jan 2021

We thank Referee #3 for their comments and suggestions, which are addressed point by point below.

Comment 1: One aspect I found to be missing was a discussion of the uncertainties in both the proxies and the measurements. $H_2SO_4$ measurements in general have associated uncertainties on the order of +/- 35-45%, however in this work none were reported. The parameters used in the proxy calculations also have associated uncertainties resulting in an overall uncertainty in the estimated $H_2SO_4$ concentration. These uncertainties (both the measurements and calculations) need to be discussed in terms of the comparisons.

Reply 1: The uncertainty associated to $[H_2SO_4]$ measurements was briefly evoked in Sect. 2.3, in the comparison between OCTAVE and STRAP datasets, and is now further discussed, in the case of STRAP, at the end of Sect. 2.2:

*"The good correlation obtained between the signals of the well characterized CI-APi-TOF and the AI-APi-TOF during the calibration experiments can undoubtedly be seen as an indicator of the satisfactory performance of the newly developed inlet, and further on the derivation of $[H_2SO_4]$ (see Fig. S3 in the Supplement of Sayhoun et al. 2019). However, it cannot be excluded that $[H_2SO_4]$ inferred from the measurements carried out during the STRAP campaign were subject to greater uncertainty due to the specific conditions of the volcanic plume, in particular with respect to $H_2SO_4$ concentrations, which were on average slightly higher in the plume than in the simulation chamber ($< \sim 5 \times 10^7$ cm$^{-3}$ in the CLOUD chamber vs $\sim 1.6 \times 10^8$ cm$^{-3}$ on average during the flight segment of interest, see Table 1 and Fig. 2)."*

Correspondingly, additional information regarding the calibration conditions during OCTAVE is now provided at the end of Sect. 2.1:

*"Note that the mass spectrometer was calibrated onsite, in the exact position it was sampling the ambient air during the measurement campaign, and up to the high sulfuric acid concentrations observed under the plume conditions."*

It is true that the effects of measurement uncertainties of both $[H_2SO_4]$ and predictor variables on the fitting procedure were not assessed in the original version of the manuscript. In order to address this aspect, a bootstrap procedure was applied to the data, following Dada et al. (2020), and systematic errors on the different variables were simulated in the bootstrap resamples (10 000 in total). Similar to Dada et al. (2020), we only accounted for the error related to measurement accuracy (as opposed to measurement precision) and we assumed:
- a factor of 2 uncertainty for $[H_2SO_4]$, based on the work of Kürten et al. (2012);
- 5% uncertainty in the measurement of RH and global radiation, based on manufacturer's specifications;
- 15% uncertainty in the measurement of $SO_2$ mixing ratio, based on calibration data;
- 20% uncertainty in the CS evaluation, similar to Dada et al. (2020).
For each proxy, the fitting parameters previously obtained from the original dataset were kept as a reference for deriving $[H_2SO_4]$, and the results obtained from the bootstrapped data were used to evaluate the variability ($25^{th} - 75^{th}$ percentile) of the fitting parameters and performance indicators associated to each proxy. This additional procedure is described in Sect. 3 of the revised manuscript:

*"The data were in addition submitted to bootstrap resampling to evaluate the effect of a possible systematic error related to the measurement accuracy of $[H_2SO_4]$ and predictor variables on the fitting parameters and performance indicators (i.e. R, RE and SSR). The method is described in detail in Dada et al. (2020) and is only briefly recalled here. 10 000 bootstrap resamples were generated from the original dataset by randomly replacing an existing data point with another, and the resulting time series were further multiplied by a set of random factors to simulate the presence of independent systematic errors on the different variables. For each variable, these factors (one per bootstrap sample, i.e. 10 000 in total) were drawn from a uniform distribution (in logarithmic scale) of possible biases in their respective uncertainty range. Specifically, uncertainties in the range between -50% and 100% were considered for measured $[H_2SO_4]$ (i.e. multiplying factors for $[H_2SO_4]$ in the bootstrap resamples were between 0.5 and 2) following the work of Kürten et al. (2012). According to calibration data, we*

*assumed an uncertainty of 15% in the measurement of SO₂ mixing ratio and, similar to Dada et al. (2020), we assumed 20% uncertainty in the CS evaluation. An uncertainty of 5% in the measurement of the remaining variables of interest (i.e. RH and global radiation) was finally accounted for based on manufacturer's specifications. For each function listed in Table 2, the fitting procedure was first applied to the original dataset to obtain a set of reference parameters for deriving [H₂SO₄]. The variability of the fitting parameters and performance indicators was then evaluated for each proxy by repeating the same procedure on the bootstrap resamples."*

Corresponding results are presented in Table 3, which is introduced at the beginning of Sect. 4.1:
*"The following discussion focuses on the fitting parameters and performance indicators (i.e. R, RE and SSR) obtained for the original dataset, but Table 3 presents as well an estimate of their variability (25th and 75th percentiles) inferred from the bootstrap procedure introduced in the previous section."*

As specified in Sect. 4.2, STRAP data were in contrast not submitted to bootstrap resampling in the development of the proxy F2':
"*Note that the evaluation of a possible systematic error related to the measurement accuracy of [H₂SO₄] and predictor variables on the fitting parameters and performance indicators was left behind this last test, which purpose was simply to get an estimate of the improvement in proxy performance related to the derivation of location specific coefficients.*"

Reference: Kürten, A., Rondo, L., Ehrhart, S., and Curtius, J.: Calibration of a Chemical Ionization Mass Spectrometer for the Measurement of Gaseous Sulfuric Acid, J. Phys. Chem. A, 116, 6375–6386, https://doi.org/10.1021/jp212123n, 2012.

Comment 2: Also at such high $H_2SO_4$ concentrations (approaching $10e^9$ molecule cm-3), the reagent ion will be depleted in the $NO_3$ CIMS measurements. The stated concentration calculation assumes pseudo first order kinetics which most likely are no longer applicable under these conditions. Were calibrations performed using these large $H_2SO_4$ concentrations to prove the validity of the calculation?

Reply 2: There is no inherent need of reagent $NO_3^-$ to be depleted with such a sulfuric acid concentration, and such a depletion was also not observed during the plume measurements (see Fig. R1 below). The sufficiency of reagent ion production ultimately depends on (i) the availability of reagent ions (= primary ion yield in the ionization source), (ii) the concentration to be measured, and (iii) the sensitivity to the target compound. If the $10^9$ cm$^{-3}$ target concentration would be a limit for smooth $NO_3^-$ ionization operation, then $NO_3^-$ technique could not be used to measure HOMs and other oxidized organic compounds often present in higher concentrations, and for which it is routinely applied for in field campaigns. Furthermore, as now explicitly stated at the end of Sect. 2.1, the mass spectrometer was calibrated on-site in the exact position it was also sampling the ambient air during the campaign, up to the high sulfuric acid concentrations observed during the plume measurements. The $H_2SO_4$ produced by the calibrator was simulated as usually by the method described in Kürten et al., (2012) which considers the present ambient conditions influencing, for example, the collision frequency.

[Figure]

Fig. R1: Measured raw ion signals at 10 minute averaging time for the reagent ions ($NO_3^-$ and $HNO_3 \cdot NO_3^-$) and product ions ($HSO_4^-$, $H2SO_4 \cdot NO3^-$, $H_2SO_4 \cdot HSO_4^-$ and $(H_2SO_4)_2 \cdot HSO_4^-$) during the volcanic plume.

Comment 3: Another aspect was the discussion of the airborne measurements. The best agreement between proxy and measurements was observed when no CS term was included (not very realistic). It was also stated that there were gaps in radiation data caused by improper measurements during turns, when the aircraft itself affected the amount of radiation reaching the sensor. Was upwelling radiation considered in these calculations? Assuming these measurements were made inside the plume with particles present, the nadir or reflected radiation could approach that of the zenith or incoming. An increase in the $H_2SO_4$ production term could balance the inclusion of the CS term.

Reply 3: Upwelling radiation was not included in the calculations. However, as shown in Fig. R2 below, upwelling radiation was relatively constant over the investigated period, and represented on average less than 10% of the downward radiation used in the calculations. Therefore, we do not believe that ignoring this term can explain our observations, for which other hypotheses are proposed, including different size ranges for the calculation of CS (P19, L5-7) and possibly contrasting origin of the particles contributing to the CS (P19, L8-11) in the two datasets.

[Figure]

Fig. R2: Downwelling and upwelling radiation measured onboard the French ATR-42 research aircraft during the first part of flight ETNA 13 (STRAP). Gaps in the time series were caused by improper measurements during turns.

Comment 4: I would find a statement about the estimated lifetime of gas phase $H_2SO_4$ under these high $H_2SO_4$ conditions useful too.

Reply 4: An accurate evaluation of the lifetime of gas phase $H_2SO_4$ would require a detailed knowledge of the individual sources and sinks of this particular species under the conditions of the volcanic plume. Such an investigation is beyond the scope of the present work and is not in any case needed for the proxy derivation. Also, it would add unnecessary speculation to the current narrative, which is based solely on the measured "simple" quantities. However, the first-order CS values derived from the DMPS data serve as a measure of the lifetime for aloft species which irreversibly condense by interaction with a surface (i.e., it is a measure of how fast vapours are lost on pre-existing particles), and its inverse value can be used as a first estimate of $H_2SO_4$ lifetime under these conditions.

*Specific comments and suggested grammatical changes:*

Comment 1: One large one to mention here is the omission of the proxy equation (Eq. 11) from Mikkonen 2011. It is useful and relevant to this work and not proper to ask the reader to look up.

Reply 1: The proxy developed by Mikkonen et al. (2011) is now recalled at the end of Sect. 4.1.

Comment 2: All Fig. X should be Figure X in text

Reply 2: We are not sure about the Reviewer's expectations here, but it seems that the use of the abbreviation "Fig. X" is part of ACP usage, with the exception of the beginning of a sentence, where the use of the full expression "Figure X" is requested.

Comment 3: Consider replacing "regular conditions" with "low SO2 conditions".

Reply 3: We believe that the term "low" in the proposed expression can be considered subjective, and would therefore prefer to keep the original expression, which meaning is clarified at first use (P6, L16-17: "*i.e. outside of the volcanic plume*").

Comment 4: Page 6, line 19: with had a -> which had a

Reply 4: typo corrected, thank you for noticing!

Comment 5: Page 6, line 27: in and off-plume -> in and out of plume

Reply 5: changed

Comment 6: Lower median radiation during STRAP than OCTAVE due to diurnal observations during OCTAVE. Why not compare same time of day?

Reply 6: We do not believe that the addition of such "detailed" comparison is necessary since the purpose of Sect. 2.3 is to provide an overview of the conditions encountered during the two campaigns and to highlight the specificities of each of the datasets that possibly impact the use of these data for the development of the proxies (e.g. P12, L7-8: "*the variability of the key variables driving $H_2SO_4$ production was too limited in the STRAP dataset to retrieve a realistic picture of the role of these variables in predicting [$H_2SO_4$]*") and are also to be taken into account in the interpretation of the results.

Comment 7: Page 7, line 1: in and off-plume -> in and out of plume (you use outside plume in line 3)

Reply 7: changed

Comment 8: Figure 1 - Why not use black for radiation similar to other plots. Yellow for radiation is hard to see. Consider changing the colour.

Reply 8: Changing the colour of radiation data was indeed a good suggestion to improve the readability of Figs. 1 and 2!

Comment 9: Page 10, lines 1-3: Equation is busy and unnecessary

Reply 9: Consistent with Comments 1 and 25, we think it is indeed relevant to remind the reader of the equations useful for this study. In this approach, we believe that, like Mikkonen's proxy, the expression of the temperature-dependant reaction rate between $SO_2$ and OH is of interest here since it is used in the expression of each of the proxies; it was nonetheless moved to the Supplement.

Comment 10: Page 10, line 10: define k'

Reply 10: We had indeed omitted to mention the meaning of k', which is now clearly indicated after Eq. 6 (now Eq. 5): "*where $k'$ corresponds to the multiplication of $k$ by a factor (to be determined in the fitting procedure) which partly takes into account the use of global radiation instead of [OH]*".

Comment 11: Page 10, Line 11: we defined in parallel proxy A1 -> we defined a parallel proxy A1

Reply 11: commas were added instead: "*we defined, in parallel, proxy A1*"

Comment 12: Page 10, line 18: CS was first removed in proxies F2 and A2, and it was reintroduced. . . -> the CS was first removed in proxies F2 and A2, and re-introduced. . .

Reply 12: changed

Comment 13: Page 10, line 22: respectively for "F" and "A" proxies. -> respectively for the "F" and "A" proxies.

Reply 13: changed

Comment 14: Page 10, line 24: contribute up to ∼35% to -> contribute up to ∼35% of

Reply 14: changed

Comment 15: Page 10, lines 29-30: definitions of alpha and beta need to come earlier.

Reply 15: We are not sure we understand the expectations of the Reviewer, since alpha and beta are defined right after the equation of proxy S1, in which they are used.

Comment 16: Page 11, line 13: which is defined as follow for -> which is defined as follows for

Reply 16: changed

Comment 17: Page 11 Table 2: parameters need to be defined, especially k. Is K x k = k' in equation 6? As general rule the reader should not have to read the text to understand a figure or table.

Reply 17: As we have now indicated (see Reply 10), $k'$ reflects the presence, in addition to $k$, of a factor that partially takes into account the use of global radiation instead of [OH] in the determination of [$H_2SO_4$]. As indicated in Sect. 3 (P10, L21-22), this "scaling factor" is indeed included in the pre-factors $K$ (for proxies F1-F3) and $a$ (for proxies A1-A4), and in parameter α for S1. This information has been added in the legend of Table 2, together with the definition of $k$:

"*Table 2 Proxy functions. F1-F3 are the proxies with powers fixed to -1 or 1 for all variables, as predicted by the theory, while A1-A4 have individual adjusted powers for each variable. S1 includes the additional $H_2SO_4$ sink related to cluster formation.* In each of the proxies, $k$ corresponds to the temperature dependant reaction rate between $SO_2$ and $OH$. *Fitting parameters K in F1-F3, $a - f$ in A1-A4 and α - β in S1 were determined iteratively to minimise the sum of squared residuals associated to each proxy.* The pre-factors a and K as well as parameter α are assumed to take into account the use of global radiation instead of [OH] in the different proxies.*"

Comment 18: Page 12, line 12: in a different volcanic plume, in which the conditions however overall resembled the average conditions encountered during OCTAVE. -> in a different volcanic plume with conditions similar to those encountered during OCTAVE.

Reply 18: changed

Comment 19: Page 12, line 13: and we believe that their behaviour was not well caught due to their limited number. -> and due to the limited number of measurements probably do not represent H2SO4 concentrations under such large concentrations of SO2 as a whole.

Reply 19 : changed

Comment 20: Page 12, line 30: measurement -> measurements

Reply 20: changed

Comment 21: Page 13, line 1: make them better -> improve results

Reply 21: changed

Comment 22: Page 13, line 6: in regular conditions, who noticed a better performance of the proxies when taking RH into account. -> who noticed a better performance of the proxies when taking RH into account under regular conditions.

Reply 22: changed

Comment 23: Page 13, line 22: point up that overall -> point out that the overall

Reply 23: changed

Comment 24: Page 13, line 23: shown on -> shown in

Reply 24: changed

Comment 25: Page 14, line 23: Eq. (11). At first look I thought this referred to Eq. (11) in the present work, which doesn't exist. I'm assuming this refers to Eq. (11) in Mikkonen et al. (2011). The equation needs to be added here so the reader doesn't have to go look it up!

Reply 25: As already mentioned in Reply 1 above, the proxy developed by Mikkonen et al. (2011) is now explicitly recalled at the end of Sect. 4.1.

Comment 26: Page 19, line 5: CS -> CSs

Reply 26: changed

Comment 27: Page 23, line 15: The condensation sink (CS) was in a first approach considered as the only sink contributing to the balance of [$H_2SO_4$] to derive seven proxies based on the knowledge of $SO_2$ mixing ratios, global radiation, CS and RH, which inclusion in the sink term was tested in several formulations. -> The condensation sink (CS) was in a first approach considered as the only sink contributing to the balance of [H2SO4] to derive seven proxies based on the knowledge of SO2 mixing ratios, global radiation, and CS. RH, included in the sink term was tested in several formulations.

Reply 27: changed

Comment 28: Page 23, line 28: In contrast, the inclusion of RH,. . . This sentence is long and runs on. Needs to be rewritten and split up to convey conclusions.

Reply 28: As suggested, the sentence was split into three shorter sentences: *"In contrast, the inclusion of RH, either with the CS or with a separate power, did not improve the performance of proxies A3 and A4, respectively, compared to A1. This observation suggested a limited effect of RH on the sink regulating $H_2SO_4$ production, in spite of the relatively high negative correlation observed between these two variables. More importantly, power -1 attributed to RH in proxy F3 lead to a systematic underestimation of [$H_2SO_4$]."*

Comment 29: Page 24, line 1: and on average higher at – but was on average higher at

Reply 29: changed

Comment 30: Page 24, line 4: the literature -> that work

Reply 30: changed

---

## Author Comment (AC2) · 27 Jan 2021

We thank Referee #2 for their comments and suggestions, which are addressed point by point below.

Review of Rose et al. Sulfuric acid proxies. This paper tries out various parameterizations for sulfuric acid in the high $SO_2$ conditions of a volcanic plume. **A.** Motivation for this work seems thinly presented, e.g. is an aim to provide a simple equation for GCMs? **B.** One set of field data was simulated somewhat reasonably but the other was not. Is that a sign that these proxies are limited in their application? **C.** A third finding (p.23 line 11) could be the limited scope of SA proxies and thus a chemical model (e.g. Boy 2005) will be needed in lieu of widespread measurements of SA.

**A.** Addressing the lack of [$H_2SO_4$] measurements, which rely on the use of state-of-the-art instrumentation that requires specific expertise, is presented in the abstract (P1, L21-23) and in the introduction (P3, L1-4) as the primary motivation for the development of the proxies. The recent multiplication of studies dedicated to the development of such proxies listed in the introduction illustrates the interest of such tools (e.g. Lu et al., 2019; Dada et al., 2020), which are not necessarily chemically detailed, but which primary objective is, again, to compensate for the absence of measurements. As highlighted in the abstract (P1, L23) and now clearly indicated in the introduction as well, the knowledge of [$H_2SO_4$], and in particular the possibility of obtaining an estimate of this quantity from more commonly measured variables, is of particular interest for the study of nucleation and NPF processes:

"*However, as recently noticed by Lu et al. (2019), direct measurements of [$H_2SO_4$] remain challenging, because the deployment of CIMS and the analysis of the data they provide require specific expertise. Therefore, for studies in which [$H_2SO_4$] is an important variable (i.e. mainly for nucleation and NPF studies), is it useful to be able to predict it from more accessible observations such as $SO_2$ concentration and environmental parameters. This is why several proxies for [$H_2SO_4$] have been developed, based on the assumption that $H_2SO_4$ mostly results from the oxidation of the sulphur dioxide ($SO_2$) by the hydroxyl radical (OH).*"

This is particularly the case in volcanic plumes, where sulfuric acid is expected to play an important role in the process (P2, L18-19, Sahyoun et al., 2019); as indicated at the end of the introduction (P3, L25-28; rephrased in the revised version of the manuscript, see Reply 6), the absence of proxies dedicated to this specific environment was in the end the main reason for this study. Motivations related to the specificities of the volcanic plume conditions are also outlined in the abstract (P1, L24-27) and recalled at the end of Sect. 2.3 (P7, L10-12).

**B.** There is indeed a limit to the application of a given proxy, since it is built on the basis of data from a specific environment, and is therefore not necessarily suitable for the description of other conditions. This limitation is clearly evoked several times in the manuscript: in the abstract (P2, L7-10), in the results section (P20, L11-13) and in the conclusion (P24, L21-25). This limitation is also more widely illustrated by the recent multiplication of studies dedicated to the construction of proxies adapted to contrasting environments (Lu et al., 2019; Dada et al., 2020). Also, the development of proxies is obviously based on a compromise between simplicity and accuracy, but in view of the results obtained (both in this work and in the literature), this approach seems to lead to satisfactory results in various conditions.

**C.** Based on Boy et al. (2005), it seems that the use of such chemical model requires the knowledge of a certain number of variables, as indicated for instance in the abstract of the abovementioned paper: "Sulphuric acid concentrations were measured and calculated based on pseudo steady state model with corresponding measurements of CO, NOx, $O_3$, $SO_2$, methane and non-methane hydrocarbon (NMHC) concentrations as well as solar spectral irradiance and particle number concentrations with size distributions". Some of the listed species (in particular NMHC) are not routinely measured continuously, and therefore we would prefer to avoid considering such compounds in our study. Our willingness to use a limited number of commonly measured variables is indicated several times in the paper (P10, L7-9; P11, L5-9). More broadly, detailed

chemical investigation and/or description of the formation pathways of $H_2SO_4$ and its precursors in a volcanic plume is behind the scope of the present work, as now further specified in Sect. 3: "*However, since we did not observe a very specific behaviour of these species in the plume compared to regular conditions which could have motivated their inclusion, we rather chose to limit the number of variables to get as simple as possible expressions for the proxies. Similarly, the dependence of $H_2SO_4$ production term over absolute water concentration was left behind from the present work in order to avoid over-constraints which could prevent the use of the newly developed proxies in datasets collected in different volcanic plumes. More broadly, while Dada et al. (2020) explicitly aimed at understanding the different mechanisms of sulfuric acid formation and losses in different environments, detailed chemical investigation and/or description of the formation pathways of $H_2SO_4$ and its precursors in a volcanic plume was behind the scope of the present work, which objective was, again, to obtain the simplest possible description of [$H_2SO_4$] from a limited set of commonly measured variables.*"

However, if the Referee is interested in conducting a more detailed analysis of $H_2SO_4$ chemistry in a volcanic plume using a model such as the one used in Boy et al. (2005), we would be pleased to contribute with the measurements we have.

Some detailed comments

Comment 1: P1, L20: 'all variables equal weight'? Do F1, F2, and F3 represent just cases of different fitted forms? 'adjusted powers' is this varying weights in the fits or also the fitted forms? Succint description of the main analysis in the abstract should not mislead a potential reader...

Reply 1: A sentence was added to help clarifying this point: "*A specific combination of some or all of these variables was tested in each of the seven proxies. In three of them (F1-F3), all considered variables were given equal weight in the prediction of [$H_2SO_4$], while adjusted powers were allowed (and determined during the fitting procedure) for the different variables in the other four proxies (A1-A4)*".

Comment 2: P1, L25: But it seems S1 did not do well for the second case?

Reply 2: It seems that the reviewer is referring to L35 instead. However, we are not sure we understand the comment, since L35 concerns the analysis of the performance of the proxies in the first case study only, while the second case is discussed right after (from P1, L35 to P2, L5). The purpose here is to summarize the results obtained for each dataset from the elements that seem most relevant, which we believe does not include the performance of proxy S1 in the case of the STRAP data.

Comment 3: P2, L21-24: This is a mangled sentence that needs to be parsed out or eliminated. The next sentence begins a new thought and deserves a new paragraph.

Reply 3: The sentence has been rephrased, and new reference was introduced:
"*Information about the species contributing to cluster formation with sulphuric acid and preferential formation pathways was gained from laboratory studies (Hanson et al., 2002, 2006). Laboratory experiments have also made it possible to evaluate instrumental setups and related protocols for accurate detection and quantification of the clusters and their precursors (Jen et al., 2016; Riva et al., 2019).*"

On the other hand we have kept only one paragraph, since all the sentences that it contains concern the detection of the clusters and their precursors.

Reference: Riva, M., Rantala, P., Krechmer, J. E., Peräkylä, O., Zhang, Y., Heikkinen, L., Garmash, O., Yan, C., Kulmala, M., Worsnop, D., and Ehn, M.: Evaluating the performance of five different chemical ionization techniques for detecting gaseous oxygenated organic species, Atmos. Meas. Tech., 12, 2403–2421, https://doi.org/10.5194/amt-12-2403-2019, 2019.

Comment 4: P3, L16: So a main motivation was to duplicate the Lu et al. study? Needs rewording.

Reply 4: In case there would be an ambiguity related to the use of the expression "*this work*", this expression has been replaced by "*the study of Lu and co-workers*".

Comment 5: P3, L22 : Points out a flaw in the proxy idea in general: missing information is important such as NO, $NO_2$, $CH_2O$, $O_3$, radiative environment, aerosol hygroscopicity?

Reply 5: As indicated in the paper and recalled in the reply to part A. of the main comment, the objective of the proxies is to provide an estimate of $[H_2SO_4]$ in absence of direct measurements. While some proxies do include some more specific variables (e.g. alkenes concentration in Dada et al., 2020, [HONO] in Lu et al., 2019), most of them (including those developed in the present work) are based on a limited number of commonly measured variables to maximize their usefulness (see reply to part C. of the main comment).

Comment 6: P3, L25: This is an unusual statement: they used the proxies but did not test them? Do you mean compared to measured H2SO4? Did not Rose 2019 have measurements?

Reply 6: As indicated on P3, L28, the present work reports "*the first direct measurements of $[H_2SO_4]$ conducted in plume conditions*", implying that, in fact, measurements were not available in Rose et al. (2019) nor in Boulon et al. (2011). This is now clearly stated:
*"In absence of direct measurements, and also of a specific proxy, Boulon et al. (2011) and Rose et al. (2019) […] and the Piton de la Fournaise, respectively. However, the lack of measured $[H_2SO_4]$ obviously did not make it possible in these studies to assess the performance of the abovementioned proxies in such unusual conditions, which has motivated the present work".*

Comment 7: P4, L15:. 'compared to' -> 'than'. Nonetheless, why care about the previous one?

Reply 7: The reader is referred to earlier studies for a more detailed description of the instrumental setup, it seems therefore interesting to mention the changes made to this setup.

Comment 8: P4, L16: One has $O_3$ and NOx too... seems that a Boy 2005 model simulation could be done.

Reply 8: The Reviewer is referred to part C. of the reply to the main comment.

Comment 9: P4, L24: made a non-negligible contribution to what ? Read on... Is there ion molecule clustering going on or are these ionization of neutral clusters? Should 195u be multiplied by 2 and 293u by 3 to get the total SA being produced? How does one separate ionization of H2SO4 dimer and trimer from the clustering of HSO4- with H2SO4? Still no reference for this equation. Or for the AI instrument, next page. Was the calibrations carried out to test the limits of these equations?

Reply 9: We first would like to mention that we noticed a mistake in the numerator of Eq. 1 which has been corrected, so that Eq. 1 now reads:

$$[H_2SO_4] = \frac{HSO_4^- + H_2SO_4 \cdot HSO_4^- + (H_2SO_4)_2 \cdot HSO_4^- + H_2SO_4 \cdot NO_3^-}{NO_3^- + HNO_3 \cdot NO_3^- + (HNO_3)_2 \cdot NO_3^-} \times C$$

Also the reference associated to this equation was not correct, it should have been Kürten et al. (2012) instead of 2014.

As now explicitly stated at the end of Sect. 2.1, the mass spectrometer in OCTAVE campaign was calibrated on-site, in the exact position it was sampling the ambient air during the measurement campaign, and up to the high sulfuric acid concentrations observed under the plume conditions. The $H_2SO_4$ produced by the calibrator was simulated as usually according to Kürten et al. (2012), which considers the present ambient conditions influencing, for example, the collision frequency. Thus, in this setup the instrumental influence was minimized, as it is indeed not straightforward to answer which of

the clusters is formed first. Yet, as shown in Fig. R1, the H₂SO₄·HSO₄⁻ signal was always minor in comparison to HSO₄⁻ and H₂SO₄·NO₃⁻ signals, and the (H₂SO₄)₂·HSO₄⁻ far smaller than those.

[Figure]

Fig. R1: Measured raw ion signals at 10 minute averaging time for the reagent ions (NO₃⁻ and HNO₃·NO₃⁻) and product ions (HSO₄⁻, H2SO4·NO3⁻, H₂SO₄·HSO₄⁻ and (H₂SO₄)₂·HSO₄⁻) during the volcanic plume.

With regard to the setup used during STRAP, although it was previously described in detail in Sahyoun et al. (2019), certain indications should indeed have been recalled in this work, both concerning the instrument itself and the calibration procedure. Those have been added to Sect. 2.2, with, as suggested, a short discussion on the possible effects related to calibration conditions different from those encountered during the campaign on the estimation of [H₂SO₄]:

"*As previously explained in details by Sahyoun et al. (2019), sulfuric acid concentrations were measured with an APi-TOF equipped with an ambient ionization (AI) inlet adapted to airborne measurements and used for the first time during STRAP. In contrast with the CI inlet, the AI inlet does not require the use of any chemicals, and only includes a soft X-ray source (Hamamatsu L9490) to ionize the sample flow. This direct ionization process was sufficient to get a high enough signal and allow a time resolution as high as 1s for the corresponding measurements. Also, in order to avoid possible effects related to pressure changes on the detection of the AI-Api-TOF, a pressure stabilizing unit was installed in front of the instrument. As detailed in Sahyoun et al. (2019), calibration of this new setup was performed (with respect to [H₂SO₄] measurement) during fall 2016 at the CLOUD CERN facility (Kirkby et al., 2011; Duplissy et al., 2016 and references therein) by comparison with the measurements performed with a nitrate based CI-APi-TOF in various conditions representative of the atmosphere. During these experiments, O₂⁻ was assumed to be the main ionizing agent of H₂SO₄, as on board the aircraft during the measurement campaign, but contribution of NO₃⁻ could not be excluded, in particular in presence of higher NOx levels (up to 33 ppb) in the CLOUD chamber. Therefore, estimates of [H₂SO₄] were finally obtained by the mean of Eq. (2) using the signals measured at m/z = 97 Th (HSO₄⁻) and m/z = 160 Th (NO₃⁻ · H₂SO₄) by the AI-APi-TOF and a calibration coefficient C = 4.5 ×10⁹ molecule cm⁻³:*

$$[H_2SO_4] = \frac{HSO_4^- + NO_3^- \cdot H_2SO_4}{Total\ ion\ count} \times C \qquad (2)$$

*The good correlation obtained between the signals of the well characterized CI-APi-TOF and the AI-APi-TOF during the calibration experiments can undoubtedly be seen as an indicator of the satisfactory performance of the newly developed inlet, and further on the derivation of [H₂SO₄] (see Fig. S3 in the Supplement of Sayhoun et al. 2019). However, it cannot be excluded that [H₂SO₄] inferred from the measurements carried out during the STRAP campaign were subject to greater uncertainty due to the specific conditions of the volcanic plume, in particular with respect to H₂SO₄ concentrations, which were on average slightly higher in the plume than in the simulation chamber ($< \sim 5 \times 10^7$ cm$^{-3}$ in the CLOUD chamber vs $\sim 1.6 \times 10^8$ cm$^{-3}$ on average during the flight segment of interest, see Table 1 and Fig. 2)".*

Reference: Kürten, A., Rondo, L., Ehrhart, S., and Curtius, J.: Calibration of a Chemical Ionization Mass Spectrometer for the Measurement of Gaseous Sulfuric Acid, J. Phys. Chem. A, 116, 6375–6386, https://doi.org/10.1021/jp212123n, 2012

Comment 10: P5, L30-32: These two sentences add nothing to the paper, instead they distract. Instead here is where some discussion of the uncertainties in the H2SO4 measurements and whether the particulars of the deployment(s) have affected them. Did exp. conditions perhaps introduce more factors (e.g. ambient pressure dependencies of the C factors)?

Reply 10: As suggested by the Reviewer, the two abovementioned sentences were removed.

The Reviewer is referred to Reply 9 with regard to the possible effects of the particular conditions of deployment of the instruments on the [H₂SO₄] measurements.

Also, a complementary investigation on the effect of a possible systematic error related to the measurement accuracy of [H₂SO₄] and predictor variables on the fitting parameters and performance indicators (i.e. R, RE and SSR) of the newly developed proxies was added at the end of Sect. 3:
"*The data were in addition submitted to bootstrap resampling to evaluate the effect of a possible systematic error related to the measurement accuracy of [H₂SO₄] and predictor variables on the fitting parameters and performance indicators (i.e. R, RE and SSR). The method is described in detail in Dada et al. (2020) and is only briefly recalled here. 10 000 bootstrap resamples were generated from the original dataset by randomly replacing an existing data point with another, and the resulting time series were further multiplied by a set of random factors to simulate the presence of independent systematic errors on the different variables. For each variable, these factors (one per bootstrap sample, i.e. 10 000 in total) were drawn from a uniform distribution (in logarithmic scale) of possible biases in their respective uncertainty range. Specifically, uncertainties in the range between -50% and 100% were considered for measured [H₂SO₄] (i.e. multiplying factors for [H₂SO₄] in the bootstrap resamples were between 0.5 and 2) following the work of Kürten et al. (2012). According to calibration data, we assumed an uncertainty of 15% in the measurement of SO₂ mixing ratio and, similar to Dada et al. (2020), we assumed 20% uncertainty in the CS evaluation. An uncertainty of 5% in the measurement of the remaining variables of interest (i.e. RH and global radiation) was finally accounted for based on manufacturer's specifications. For each function listed in Table 2, the fitting procedure was first applied to the original dataset to obtain a set of reference parameters for deriving [H₂SO₄]. The variability of the fitting parameters and performance indicators was then evaluated for each proxy by repeating the same procedure on the bootstrap resamples.*"

Corresponding results are presented in Table 3, which is introduced at the beginning of Sect. 4.1:
"*The following discussion focuses on the fitting parameters and performance indicators (i.e. R, RE and SSR) obtained for the original dataset, but Table 3 presents as well an estimate of their variability (25ᵗʰ and 75ᵗʰ percentiles) inferred from the bootstrap procedure introduced in the previous section.*"

As specified in Sect. 4.2, STRAP data were in contrast not submitted to bootstrap resampling in the development of the proxy F2': "*Note that the evaluation of a possible systematic error related to the measurement accuracy of [H₂SO₄] and predictor variables on the fitting parameters and performance*

*indicators was left behind this last test, which purpose was simply to get an estimate of the improvement in proxy performance related to the derivation of location specific coefficients."*

**Comment 11**: P7, L3: Boy 2005 showed that NO+HO$_2$ was a major source. In fact, why not run that model in the plume for comparison?

**Reply 11**: We did not have continuous measurement of [OH] during OCTAVE, so we cannot compare model outputs with observations. Also, we tried to quantify [HO$_2$] with Br- mass spectrometer but were unsuccessful; we know it is not a simple quantity to estimate, and thus we do not want to attempt it either. More broadly, as already mentioned in the reply to the main comment, the aim of this work was not to investigate nor describe in detail the formation pathways of H$_2$SO$_4$ and its precursors in a volcanic plume, but to provide simple expressions to estimate [H$_2$SO$_4$] with a sufficient level of confidence.

**Comment 12**: P10, L8: A. This seems to be not only wrong chemically but misguided. O$_3$ times radiation (hv) is the correct parameter and you should use it as such. Let potential future users input the ozone for their situation. Also, when water is low such as cold temperatures at altitude, quenching of O(1D) by air molecules would compete. B. Is SO2 in the plume so high that it is optically thick? Would the UVB be affected more than global radiation (this reveals that 'global radiation' and its measurement should be explained in depth in this context)? C. Again, Boy 2005 showed that NO + HO2 was the major direct source for OH. All these questions could be addressed by some sort of OH box model run for plume conditions.

**Reply 12**:

A. Replacing [OH] with global radiation was approved in previous studies dedicated to the development of proxies for [H$_2$SO$_4$] in various environments (Petäjä et al., 2009; Mikkonen et al., 2011; Dada et al., 2020). Following the Reviewer's suggestion, we have nonetheless tried to replace global radiation by the product $Rad \times [O_3]$ in all the proxies. As now shown in the Supplement (Table S1 and Figs. S3-S4), this did not lead to any significant improvement of the predictive ability of the proxies which, with the exception of A2, all saw a deterioration of their performance indicators. This sensitivity test and the corresponding results are briefly discussed in Sect. 4.1:
   *"As a sensitivity test, an attempt to replace global radiation by the product $Rad \times [O_3]$ was made in all the proxies, to investigate if the explicit consideration of O$_3$, which photolysis is the main pathway for OH formation during daytime, would allow to further optimize the prediction of [H$_2$SO$_4$]. Corresponding results are reported in the Supplement (Table S1 and Figs. S3-S4), but do not highlight any improvement in the performance of the proxies, which, with the exception of A2, all display worse performance indicators than when considering global radiation alone. Limited improvement in the predictive ability of the proxies was also noticed by Lu et al. (2019) when considering [O$_3$] with UVB in the urban atmosphere of Beijing, where the concurrent inclusion of [HONO] seemed in contrast to be more critical."*
   Regarding the conditions, they were not particularly dry at Maïdo during the OCTAVE campaign (see Table 1 and Fig. 1, median RH ~ 68% in plume conditions), and are not in general, as reflected by the frequent presence of clouds in the vicinity of the station. Moreover, despite its altitude, this station does not experience very cold temperatures due to its geographical location (see Fig. 8 in Foucart et al., 2018, monthly averages > ~9°C in 2015). More broadly, even if high altitude sites (in particular those located at higher latitudes) do in general experience colder temperatures compared to lowland sites, they are also often associated with high frequency of cloud occurrence (e.g. 60% on average during winter at puy de Dôme, France, 1465 m a.s.l.; Baray et al., 2019), which indicates that they are not characterized by particularly dry conditions. So we do not think there is a need to consider any term in the proxies related to quenching of O(1D) by air molecules that would be justified by specific conditions at high altitude.

Reference: Baray, J.-L., Bah, A., Cacault, P., Sellegri, K., Pichon, J.-M., Deguillaume, L., Montoux, N., Noel, V., Seze, G., Gabarrot, F., Payen, G. and Duflot, V.: Cloud Occurrence Frequency at Puy de Dôme (France) Deduced from an Automatic Camera Image Analysis: Method, Validation, and Comparisons with Larger Scale Parameters, Atmosphere, 10(12), 808, doi:10.3390/atmos10120808, 2019.

B.   No, the plume was not optically thick, because $SO_2$ concentration was very high only for brief moments; the highest concentrations measured during OCTAVE (> 200 ppb) have furthermore been excluded from the analysis (P12, L16-18). It cannot be excluded, however, that the response of UVB radiation was slightly different from that of global radiation to plume conditions. However, this analysis is outside the objectives of this work, where, consistent with the abovementioned studies, the choice was made to consider global radiation, once again to favour the reuse of the proposed proxies (P10, L7-9), since UVB is generally not systematically measured.

C.   The Reviewer is referred to reply 11 as well as reply to part C. of the main comment.

Comment 13: P10, L10: kprime is what? The parameter K must have some physical meaning. Also, CS concerns here. On the face of it, the CS used is not correct and one should apply correction factors as the aerosol is probably hygroscopic (loads of $SO_2$ here.) Furthermore: below 40%? The level of drying is not even known? Uncertainties pile up. A nominally hygroscopic aerosol should be assumed and the CS should corrected as best as possible, perhaps using half the nom.hyg.aer. correction.

Reply 13: The meaning of k'is now indicated after Eq. 6 (which is now Eq. 5): "*where k′ corresponds to the multiplication of k by a factor (to be determined in the fitting procedure) which partly takes into account the use of global radiation instead of [OH]*". Some information about the meaning of K was provided on P10, L20-22, and is now recalled also in the legend of Table 2: "*The pre-factors a and K as well as parameter α are assumed to take into account the use of global radiation instead of [OH] in the different proxies.*"

Regarding the treatment of CS, we used the same approach as Mikkonen et al. (2011), who also calculated the CS in dry conditions, and did not apply a specific correction for the hygroscopic particle growth (this is also the case for the data from Budapest in Dada et al., 2020). There is no detailed characterisation of the particle hygroscopic growth factor at Maïdo, so, in line with the following lines from Mikkonen et al. (2011), we rather used the simple approach of including RH directly in some of the proxies: "Also, hygroscopic particle growth is expected to differ between different measurement sites and is, moreover, a function of season and air mass. A careful consideration of the hygroscopic growth effect on CS would thus require considerable additional effort." Still following the work of Mikkonen and co-workers, detailed consideration of the particle hygroscopicity might in addition have limited value only: "Sensitivity tests for Hyytiala data indicate that the hygroscopicity correction is not of significant magnitude to remarkably improve the calculated sulphuric acid proxies".

These aspects are now explicitly mentioned in the revised version of the paper (Sect. 3):
"*Applying a specific correction for the hygroscopic particle growth would have required a detailed characterisation of this process (e.g. as a function of air mass type, season) which has not been performed at Maïdo, and is likely not either available at a number of sites where the newly developed proxies could be used. Also, according to Mikkonen at al. (2011), such hygroscopicity correction might, at least in some environments, have only limited effect on the prediction of [H2SO4]. Therefore, similar to Mikkonen et al. (2011), inclusion of RH in the sink term was tested instead*".

Comment 14: P10, L25: That is huge,35%. It would be a dominant contribution to the signal at 195u.

Reply 14: There seems to be a misunderstanding here, because the term that is included in proxy S1 $(-\beta \times [H_2SO_4]^2)$ does not directly correspond to the signal measured at 195 amu, but to the square of the acid concentration to which all the ions present in the numerator of Eq. 1 are expected to contribute.

Comment 15: P10, L30 Give some insight on this term. It represents a net flux to the dimer. That this is a constant for a whole measurement campaign is not realistic. Agreement with any estimated dimer abundance / loss rate should be discussed.

Reply 15: As indicated in the paper (P10, L23-26), the term $-\beta \times [H_2SO_4]^2$ is introduced in S1 as a proxy for the sink of $H_2SO_4$ related to molecular cluster formation. As recalled in Reply 14, it is expressed as a second order function of the sulfuric acid concentration, which is calculated from the signals of all the ions present in Eq 1., among which the so-called dimer.

The systematic consideration of this term in the proxy may indeed be questionable, since the occurrence of molecular cluster formation/nucleation is not continuous. However, the high $H_2SO_4$ levels encountered in volcanic plumes seem to favour the occurrence of the process, as evidenced by the high NPF frequencies reported by Rose et al. (2019), thus indicating that the inclusion of this term is certainly often relevant/justified under plume conditions. Finally, the fairly good performance observed for S1 (among the best for OCTAVE with A1) confirms that considering this additional term contributes to satisfactory predictions for [$H_2SO_4$], or at least does not strongly alter them.

Comment 16: P11, L9: What ? Essentially saying: We don't want to know too much. Also arguing that proxies are not meant to be all that useful? Instead, put as much as you know into it. Then knowledge develops, limitations in it become known.

Reply 16: The Reviewer is referred to the replies to the main comment.

Comment 17: These last few comments underscore the many levers of uncertainty here. The final section has too-much-summary of the various proxies and there is not enough presentation of a firm conclusion regarding the limits of their chosen proxy (let's not have 8 or 9 to choose from) in the light of the uncertainties.

Reply 17: We hope that with the answers previously proposed and the clarifications made to the objectives (compensating for the absence of measurements, not aiming at a detailed understanding of $H_2SO_4$ formation pathways) and to the approach chosen for the development of proxies (based on a limited number of commonly measured variables), the uncertainty levers identified by the Reviewer will no longer be considered as such.

We are not sure what the Reviewer means by «The final section has too-much-summary of the various proxies". Concerning the fact that no firm indication is given regarding the proxy to select, it is true. This is explained by the fact that, overall, all suggested proxies display performances which, although slightly contrasted over the different [$H_2SO_4$] ranges and possibly variable depending on the specificities of the datasets and/or associated environments, can be globally considered acceptable. It is hardly possible, on the basis of our observations/results, to provide more directive indications. The only criterion that is not a priori subjective and that can possibly guide the choice is based on the availability, for a given site, of the data necessary for the application of the various proxies.

---

## Author Response (AR2)

Dear Editor,

Please find attached a revised version of our manuscript, which, as suggested, includes in the first part of the introduction some additional elements regarding the involvement of sulfuric acid in cluster formation processes. The different $H_2SO_4$-driven nucleation pathways investigated in laboratory studies and the participation of ions in the process, which are both related to the conditions in which the process occurs, are in particular evoked. In contrast, we have not included any detailed discussion of the thermodynamics associated with these different mechanisms, since, although the knowledge of $[H_2SO_4]$ is of particular interest for nucleation and NPF studies, understanding of these processes is not part of the objectives of this study. Similarly, we have not made any specific focus on upper tropospheric conditions, since the lower tropospheric levels are likely impacted by volcanic eruptions as well, in particular when these are taking place in the planetary boundary layer (e.g. Stromboli, 924 m a.s.l.; Sahyoun et al., 2019).

In addition, all acronyms and equations are now displayed in regular font.

Thank you for considering our revised manuscript.

Sincerely,

Clémence Rose

---

## Author Response (AR3)

Dear Editorial board,

Please not that legend of Fig. 9.b was slightly moved in the final version provided in the .zip folder in order to facilitate the reading of the data; the content of the figure is otherwise unchanged.

Sincerely,

Clémence Rose